# A facile one step route that introduces functionality to polymer powders for laser sintering

Eduards Krumins[1], Liam A. Crawford[2], David M. Rogers [1], Fabricio Machado [1,3], Vincenzo Taresco[1], Mark East [4], Samuel H. Irving[1], Harriet R. Fowler [1], Long Jiang [5], Nichola Starr[5], Christopher D. J. Parmenter[6], Kristoffer Kortsen [1], Valentina Cuzzucoli Crucitti[4], Simon V. Avery [2], Christopher J. Tuck [4] & Steven M. Howdle [1] ✉

Laser Sintering (LS) is a type of Additive Manufacturing (AM) exploiting laser processing of polymeric particles to produce 3D objects. Because of its ease of processability and thermo-physical properties, polyamide-12 (PA-12) represents ~95% of the polymeric materials used in LS. This constrains the functionality of the items produced, including limited available colours. Moreover, PA-12 objects tend to biofoul in wet environments. Therefore, a key challenge is to develop an inexpensive route to introduce desirable functionality to PA-12. We report a facile, clean, and scalable approach to modification of PA-12, exploiting supercritical carbon dioxide (scCO$_2$) and free radical polymerizations to yield functionalised PA-12 materials. These can be easily printed using commercial apparatus. We demonstrate the potential by creating coloured PA-12 materials and show that the same approach can be utilized to create anti-biofouling objects. Our approach to functionalise materials could open significant new applications for AM.

Laser Sintering (LS) or Laser Powder Bed Fusion (L-PBF) is an Additive Manufacturing (AM) process that produces components from a digital file (e.g. STL), via sequential laser processing of particle layers. The technique is popular because it provides a quick route to mechanically stable polymer components without the need for moulds or tooling and, unlike other AM processes, it has no requirement for support materials or structures[1–4]. PA-12 is the polymer of choice because it provides the right thermo-physical properties for the process[5–7], is commercially available[4,8,9] and allows the user to produce complex parts. But the final properties of the objects show limitations. PA-12 is white in appearance, hence colour must be imparted post printing either through spray-painting or vat dyeing which means that colour is only adsorbed on the surface and can easily be scratched[10]. Additionally, multiple coats of paint can cause inaccuracy in dimensions and the colouring process brings added cost and complexity. The surface properties of PA-12 are such that the materials produced are vulnerable to biofilm formation and rapid biofouling in the presence of a range of bacterial and fungal species including yeasts[4,10,11]. This limitation excludes LS objects from use as medical devices, food packaging and in wet e.g. maritime applications[11–17], as biofouling severely impairs functionality and aesthetics[13,15,18]. However, developing a completely new polymer system that could overcome these drawbacks is challenging in both technical and commercial requirements.

[1]School of Chemistry, University of Nottingham, University Park Nottingham, NG7 2RD Nottingham, UK. [2]Life Sciences, Faculty of Medicine and Health Sciences, University of Nottingham, University Park Nottingham, NG7 2RD Nottingham, UK. [3]Institute of Chemistry, University of Brasília, Campus Universitário Darcy Ribeiro, Brasília, DF 70910-900, Brazil. [4]Centre of Additive Manufacturing, Faculty of Engineering, University of Nottingham, 522 Derby Rd, Lenton, Nottingham NG7 2GX, UK. [5]School of Pharmacy, University of Nottingham, University Park Nottingham, Nottingham NG7 2RD, UK. [6]Nottingham Nanoscale and Microscale Research Centre, University Park, University of Nottingham, Nottingham NG7 2RD, UK. ✉e-mail: steve.howdle@nottingham.ac.uk

Here, we report a versatile and sustainable approach to create functional powders for LS that can be printed using currently available commercial apparatus. We have demonstrated our approach by modification of commercially sourced PA-12 powder and we describe a facile process that provides each individual particle with a thin functional polymeric coating in a single step. To do this, we exploit the unique properties of supercritical $CO_2$ (scCO$_2$), which has been used previously to synthesise and process a wide range of polymers[19–22]. The low viscosity and high diffusivity of scCO$_2$ allows for effective free radical polymerisation (FRP) of soluble monomers and excellent penetration of the growing functional polymer coating into the PA-12 sub-surface.

For the coatings, we targeted a major component that has a close match to the thermal and mechanical properties of PA-12 and a minor component that delivers functionality. The major component is poly(isobornyl methacrylate) (PIBMA) with glass transition temperature ($T_g$) ~180 °C which is very close to the melt temperature ($T_m$) of PA-12 (~175 °C) and hence, both the outer shell and the core of each particle should sinter contemporaneously maintaining the wide sintering window of PA-12[4,23]. Additionally, IBMA is commercially available, and the monomer can be obtained via renewable and sustainable reaction pathways[24].

The minor functional component (2.5–10 wt% of the coating) is introduced via radical copolymerisation with IBMA. To prepare coloured PA-12 particles we designed monomers that carry a coloured functionality. Commercial dyes were sourced that could be functionalised with a methacrylate moiety without significantly changing their visible light absorption profile. The dyes were screened via time-dependent density-functional theory (TDDFT) calculations (Supplementary Tables 1 and 2; Supplementary Fig. 4) to eliminate significant hypsochromic or bathochromic shifts[25,26]. Computed UV spectra for the azobenzene and anthraquinone hydroxy and methacrylate monomers show (Supplementary Fig. 4) that for azobenzene there is a predicted blue shift of 8 nm for the second vertical transition upon changing from hydroxy to methacrylate. For anthraquinone the blue shift for the first vertical transition is predicted to be less than 1 nm upon changing from hydroxy to methacrylate. These blue shifts are considered negligible. Through this route, three economically viable dyes were identified with promising colour profiles.

## Results and discussion

For Disperse Red 1 and Yellow 1 (Supplementary Figs. 5 and 6) simple methacrylation was achieved via methacryloyl chloride, a route that efficiently converts hydroxyl to methacrylate functionalities[27]. For Disperse Blue 3 the synthesis proved more complex because the two secondary amine functionalities could also react with methacryloyl chloride to yield significant by-products. This selectivity issue was resolved by using the enzyme Novozym 435 to achieve methacrylation by adding methyl methacrylate (Supplementary Fig. 7).

In the coating process, PA-12 particles were added to a high pressure scCO$_2$ autoclave along with initiator, IBMA and dye monomers. The coloured copolymer forms a physical coating during the polymerisation in scCO$_2$ (Fig. 1). The surfaces of the PA-12 particles act as a locus for the growing coloured copolymer chain and as this precipitates from the continuous phase scCO$_2$ it forms on the surface. But it should be noted that the amorphous regions of PA-12 are significantly swollen by the scCO$_2$ and as such the PA-12 surface is penetrated by the growing polymer chain. Thus, leading to interpenetration at the surface layers. The coating can be removed by selected solvents in which the copolymeric coating is soluble, such as chloroform or tetrahydrofuran. However, the copolymers are insoluble in water and leaching tests have been undertaken over a period of six months showing that in aqueous environments, the coating does not leach at all. In addition, it must be pointed out that through the sintering process there is significant enhancement of the interpenetration via

melting, and in the final printed parts we see no evidence of any partition or separation at the resolutions we can interrogate.

Analysis of the solubilised coatings by GPC and $^1$H NMR demonstrate that the shell is a copolymer of IBMA and dye monomer with molecular weights in the region of 40,000 Da; values which ensure a consistent $T_g$ (Supplementary Figs. 8–23). These reactions were performed in a 60 mL high pressure autoclave and also using our larger scale 1 L apparatus to demonstrate scalability a critical aspect for commercial LS powders (Supplementary Figs. 24–33).

The coated PA-12 products were retrieved from the autoclave as dry, free flowing, highly coloured PA-12 powders[28–30] that are completely free of any surfactant or solvent residues. Furthermore, scCO$_2$ is non-toxic, non-flammable, and can be recycled[28]. Under sintering, the functional particles exhibit suitable thermal properties with $T_m$ 175 °C, $T_c$ 146 °C, and $T_g$ 167–168 °C. SEM demonstrates that the coating process for the PA-12 particles is highly efficient; across multiple SEM images there are no examples of smaller acrylic-only particles and the coloured polymer coating can clearly be seen on the surface. Analysis of individual particles using FIB-SEM (Fig. 1C) shows that the surface is coated by a layer P(IBMA-Dye monomer) as the surfaces are evidently different. TOF-SIMS analysis of particles coated with P(IBMA-DR1MA) revealed that there were chemical differences (characteristic of P(IBMA-DR1MA) between the coated and uncoated PA-12 particles) reaffirming that the coating process was successful (Fig. 1D). This is reinforced by LDS analysis showing an increase in average PA-12 particle size by 2–4 μm (average 61–63 μm) (Supplementary Figs. 11, 16, and 21).

LS processing was performed using an EOS Formiga P100 printer with the dye coated particles being used to produce $5 \times 20 \times 20$ mm squares with minimal warpage (Fig. 2A). To quantify the colour, assessments by two widely used colour scales CMYK and CIELAB were used. CMYK is often used for colour mixing and CIELAB is the standard colour scale used by the International Commission on Illumination. For P(IBMA-Y1MA), CMYK (3%, 19%, 56%, 0%) and CIELAB values (85, 43, 79°) confirm an acceptable yellow. Similar outcomes were observed for P(IBMA-DB3MA) blue and P(IBMA-DR1MA) red (Fig. 2A). The surfaces of the coated versus uncoated particles are clearly different showing the efficacy of the process (Fig. 2B). Parts with complex geometries such as large overhanging angles were built with the coloured powders, for example a popular science fiction character printed with PA-12 coated with P(IBMA-Y1MA) (Fig. 2C). This is a challenging structure and demonstrates the ability to produce thin wall structures and overhanging angles of the ears and hands at a resolution that is very close to standard PA12 Laser Sintering, but with enhanced colour definition. Moreover, this colour cannot be "scratched" and persists throughout the entirety of the part. SEM analysis of the cross-sections of the printed parts demonstrated that the particles had sintered properly during LS (Fig. 2D). Time-of-Flight Secondary Ion Mass Spectrometry TOF-SIMS analysis, focusing on functional groups unique to DR1MA and IBMA, demonstrated that the coatings are at the surface of the printed parts (Fig. 2E).

For the powder modification process to be commercially viable, the cost of production must be reasonable (Supplementary Tables 3 and 4). Only a small proportion of dye monomer is required in the outer shell (2.5–10 wt% of the coating), the majority of the coating is IBMA, a commercially available product. The outer shell itself is just ca. 15 wt% of the total coated particle material. We have found that the total amount of coloured PA-12 needed in the build can be as little as 20% (with 80% uncoated PA-12) as demonstrated by our production of a popular blue building brick toy (Fig. 3C). Other 3-D objects with a range of geometries and scales were also printed using these coloured PA-12 materials. Detailed analysis revealed that resolutions of ~100 μm could be achieved and dimensional error when comparing design to finished part was less than 0.5% (Supplementary Fig. 47).

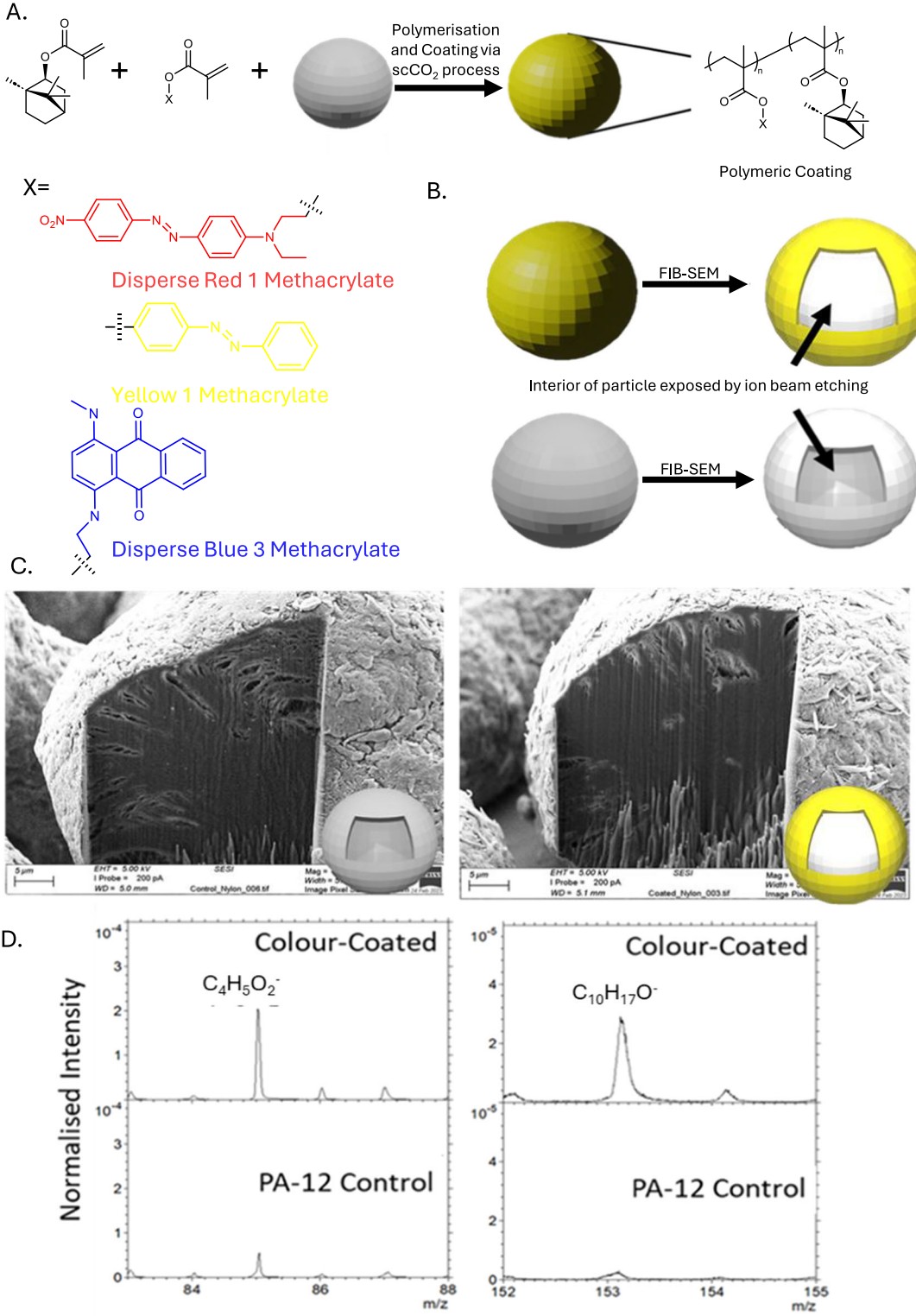

**Fig. 1 | ScCO₂ polymerisation and coating process. A** Schematic of reaction procedure: IBMA and each dye methacrylate monomer react to form a coating on each PA-12 particle. **B** Schematic of focused ion beam scanning electron microscopy (FIB-SEM) analysis on uncoated and coated PA-12 particles revealing coating and interior structures. **C** FIB-SEM images of uncoated (Left) and yellow P(IBMA-Y1MA) coated PA-12 particles (Right) showing surface coating, whilst the interior core structure is unchanged. **D** TOF-SIMS analysis of PA-12 particles coated with P(IBMA-DR1MA). Top Left: TOF-SIMS mapping of C₄H₅O₂⁻ moiety which in the particles is unique to the methacrylate component of the P(IBMA-DR1MA) coating, showing large quantities of IBMA and DR1MA on the particles. Bottom Left: showing the absence of the methacrylate group (C₄H₅O₂⁻) on the uncoated PA-12 particles. Top Right: TOF-SIMS mapping of C₁₀H₁₇O⁻ which in the particles is unique to the IBMA component, showing large quantities of IBMA on the particles. Bottom Right: showing the absence of IBMA (C₁₀H₁₇O⁻) on the uncoated PA-12 particles. For each sample, data were acquired over 3 regions of 500 μm × 500 μm at 256 × 256 pixels resolution for 20 scans. Data acquisition and analysis was performed using SurfaceLab 7 software (IONTOF GmbH).

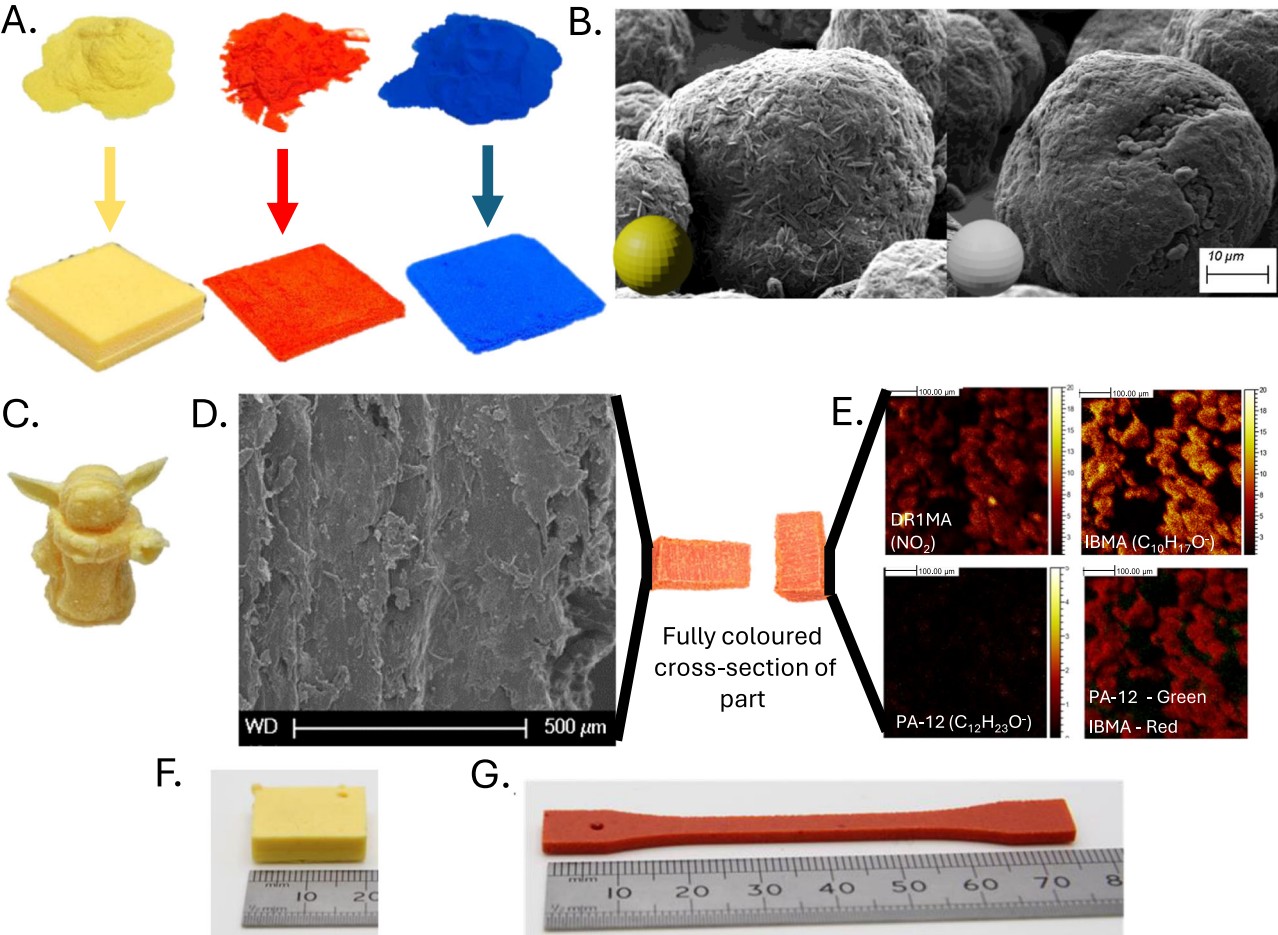

**Fig. 2 | Selective laser sintering of colour coated PA-12 particles. A** Primary coloured PA-12 particles Yellow (P(IBMA-Y1MA)), Red (P(IBMA-DR1MA)) and Blue (P(IBMA-DB3MA)). **B** SEM images of Yellow coated (P(IBMA-Y1MA) and uncoated PA-12 particles showing clear difference in surface structure. (SEM images were taken in triplicate and in varying magnifications for each sample). **C** A 3-D popular science fiction character printed from Yellow coated PA-12 particles (P(IBMA-Y1MA) showing complex geometric features such as the large overhanging ears. **D** SEM image of a cross-section of the surface of a red P(IBMA-DR1MA) printed part showing successful particle sintering. (SEM images were taken in triplicate and in varying magnifications for each sample). **E** TOF-SIMS analysis of the surface of that part Top Left: TOF-SIMS mapping of the unique $NO_2$ group showing a prevalence of red DR1MA on the printed surface. Top Right: TOF-SIMS mapping of $C_{10}H_{17}O^-$ which in the printed structure is unique to the IBMA component, showing large quantities of IBMA on the printed surface. Bottom Left: TOF-SIMS mapping of the printed surface of $C_{12}H_{23}O^-$ which is unique to the PA-12 component, indicating relatively low presence of PA-12 on the printed surface. Bottom Right: Overlayed TOF-SIMS mapping of groups unique to IBMA (red) and PA-12 (green) on surface of printed part showing prevalence of the P(IBMA-DR1MA) on the surface. For each sample, data were acquired over 3 regions of 500 μm × 500 μm at 256 × 256 pixels resolution for 20 scans. Data acquisition and analysis was performed using SurfaceLab 7 software (IONTOF GmbH). **F** Image of square composed of Yellow coated (P(IBMA-Y1MA) printed via SLS, using a ruler as a scale bar. **G** Image of tensile bar composed of Red coated (P(IBMA-DR1MA) printed via SLS, using a ruler as a scale bar.

The single-coloured PA-12 materials can be mixed physically to yield different colours giving a very flexible colour mixing system. For example, mixing yellow and blue PA12 particles leads to a green powder (Fig. 3A). Using simple physical mixing, 81 different colours were generated by varying red, yellow, blue, and white (PA-12) (Fig. 3B). The resulting mixtures were printed to yield objects ranging throughout colour space, from green to orange, brown and violet (Fig. 3B). Using this basic colour mixing we were able to target a specific blue colour via mixing white and blue PA-12 (Fig. 3C). The printed object had a colour very close to that of the target (ΔE(CMC) = 3.5) (Fig. 3C). A cross section was imaged via SEM showing uniform particle sintering of both coated and uncoated particles (Fig. 3D). Our initial studies showed that the CMYK colour mixing was not always easy to predict reliably. It is well known that predictive algorithms and other methods have been used to accurately and reliably predict the colour of formulations used in coloured inkjet printing. However, there are added complexities in a colour mixing system for SLS such as the possibility of colour change during melting, which necessitate the need for enhanced predictive methods (Supplementary Fig. 34). To address this, we developed a predictive model based on a simplex-centroid design, combined with a particle swarm optimisation (Supplementary Figs. 35–38; Supplementary Table 5). This model allows the user to predict the formulations required to obtain a target colour. Through this, we have shown that the same defined target colours can be created from different mixtures; providing a potential cost-optimisation tool to ensure manufacture using the least expensive combination of materials (Supplementary Table 6).

We also demonstrate that multicoloured objects can be produced. In this case black and white PA-12 powders were applied sequentially in a single build and a sharp interface can be observed demonstrating the potential for exploiting multicoloured powders (Supplementary Fig. 39). A key attraction of this approach is that the two differently coloured PA-12 materials will have very similar thermal properties and can be printed and processed under the same conditions, allowing for adequate binding between the different sections.

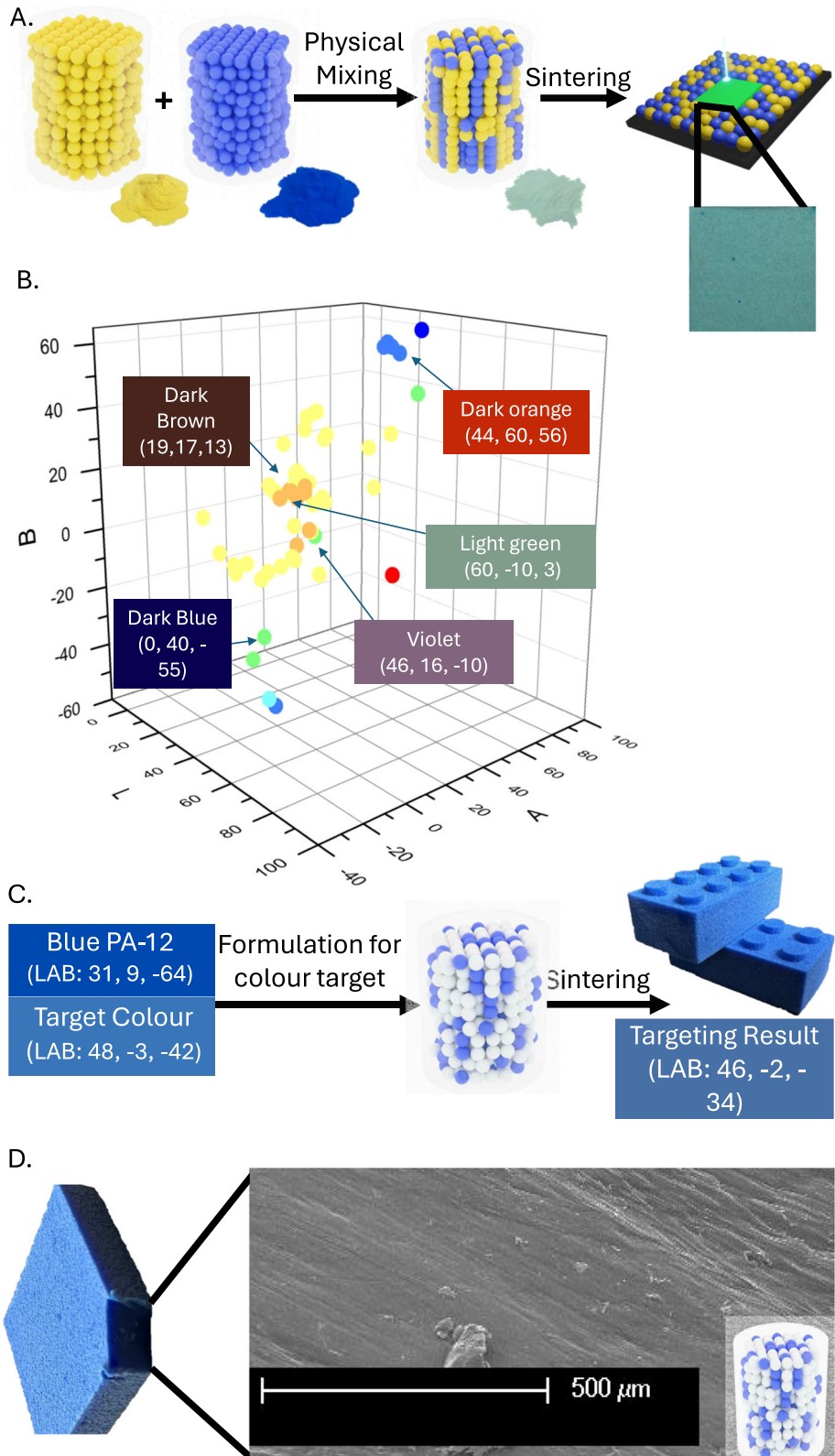

**Fig. 3 | Colour mixing (CMYK) with SLS via use of primary coloured coated PA-12 materials. A** Physical mixing of yellow and blue PA-12 materials yielding a green powder composed of unique yellow and blue particles, which when sintered form a structure with a solid green colour. **B** Graph on CIELAB colour space, showing ≥50 unique colours made through CMYK colour mixing on SLS, with 5 example colours of: Dark Blue, Dark Brown, Violet, Light Green, and Dark Orange. **C** Example of colour targeting via CMYK colour mixing. Target commercial blue colour (LAB: 48, −3, −42) was achieved through mixtures and surprisingly, only 20% of the blue powder was required to achieve a good match (LAB: 46, −2, −34) to a popular construction toy. We observed a minimal colour difference ΔE (CMC) = 3.5 (0–100 scale with 0 being exact colour match and 100 being complete opposites). **D** SEM image of cross-section of the print showing successful and uniform sintering of both materials (SEM images were taken in triplicate and in varying magnifications for each sample).

Other examples of multicoloured prints are also presented (Supplementary Figs. 47–51).

The mechanical properties of the control (virgin PA-12) and the coloured and functional PA-12 based materials were analysed utilising ISO-527-2 standards in triplicate (Supplementary Fig. 52; Supplementary Table 8). Compared to the control parts composed of virgin PA-12, a mix of commercial PA-12 and PA-12 coated with P(IBMA-DR1MA)(80/20 wt%) had a mean tensile stress at maximum load of 39.61 MPa, over 97% of that measured for the control PA-12 (Supplementary Fig. 52; Supplementary Table 8). So, we can be confident that our introduction of colour does not have a detrimental effect upon the 3D printed parts.

The range in the measured tensile stress at maximum load between the printed parts for the 80/20 mixed samples was ~2% and that of virgin PA-12 was ~1% (Supplementary Fig. 52; Supplementary Table 8). This suggests that there is further optimisation of print parameters required, and that the P(IBMA-DR1MA) is having some impact on the energy absorption characteristics of prepared material that requires further investigation.

From the mechanical property data, the coloured and functional parts have been printed successfully via laser sintering processes with minimal changes to manufacturing process parameters. In addition, extensive surface analysis revealed that the coloured PA-12 materials and the control are in the same range of values in terms of surface roughness (Supplementary Tables 9–13; Supplementary Figs. 53–55).

Having established that our scCO$_2$ coating approach works effectively to add colour, we now turn to exploiting this process to add other functionality. A longstanding drawback of PA-12 printed materials is the fact that they cannot be utilised in wet environments because of biofilm formation and biofouling. This is a significant commercial barrier to utilisation of 3-D printed objects in wet applications. Acrylic polymers based on terpenes and naturally derived acids have shown promising inhibitory properties[27,31–34] and we focussed next on a strategy for incorporating these as the functional minor component into IBMA coatings on PA-12 particles (Fig. 4A). An added advantage from a sustainability perspective is that the terpenes are naturally abundant and can also be obtained from industrial waste streams[27,31–33,35,36].

Antifouling co-polymeric coatings were synthesised at three different weight percentages (10, 20, and 30 wt%) of the functional monomer to the major component IBMA ensuring again that the observed thermal properties ($T_g$, $T_m$, and $T_c$) were suitable for LS (Supplementary Figs. 40–45; Supplementary Table 7). The samples were prepared as melt pressed discs (5 mm × 1.5 mm) of PA-12 designed to fit within the well dimensions of a 96-well microplate. An established assay of biofilm activity on surfaces using metabolic reduction was applied to the discs. A range of diverse microorganisms known to form biofilms was used to challenge the materials. These included pathogenic bacteria (*Pseudomonas aeruginosa*) and yeast (*Candida glabrata*), known to be problematic on marine filters such as reverse osmosis filters and medical devices; crop-pathogenic filamentous fungus (*Colletotrichum gloeosporioides*); and filamentous fungus (*Chaetomium globosum*) that is an industry standard for determining resistance of synthetic polymeric materials to fungi[37].

The four organisms were assayed for biofilm formation on coated versus uncoated PA-12 discs (Fig. 4B). P(IBMA-bornyl methacrylate) was the only polymer coating that significantly reduced biofilm formation of *P. aeruginosa* (Fig. 4B). The same coating was highly effective with *C. glabrata* (giving >99% inhibition of biofilm formation) while PIBMA, P(IBMA-lactic acid acrylate), P(IBMA-cinnamyl methacrylate) and P(IBMA-neryl methacrylate) also showed strong resistance to biofilm formation by this yeast (Fig. 4B). For the filamentous fungi *C. gloeosporioides* and *C. globosum* P(IBMA-oleic acid acrylate) was the most effective polymer at reducing biofilm formation (Fig. 4B). For *C. globosum* all of the coatings reduced biofilm formation significantly compared to commercial PA-12 (Fig. 4B). To support these data, discs

(5 mm × 1.5 mm) were then printed via LS from the coated PA-12 powders and tested with the same assays, which revealed similar antibiofouling properties showing that the effects are translatable (Fig. 4C).

The mechanisms underlying the formation of biofilm will depend upon surface topology, roughness, hydrophobicity and charge; the latter is particularly important for bacteria[38–42]. The effects of these surface properties are less well characterised for fungi, but the fact that the coatings were largely more effective against fungal than *P. aeruginosa* biofilms could relate to the fact that *P. aeruginosa* secretes large quantities of extracellular polysaccharides and DNA that help it adhere to surfaces[43–45]. Furthermore, biofilm formation for filamentous fungi involves a germination step, which itself is influenced by surface properties[46]. Toxicity testing showed that these materials and printed parts are largely non-toxic to the organisms (Supplementary Fig. 46). The fact that some of these coatings do clearly suppress biofilm formation across this diverse range of test organisms with very different cell wall compositions[47] and biofilm development processes[43,44,48,49] gives confidence that such materials might confer broad-spectrum anti-biofouling properties.

We have outlined a facile approach to develop functional coatings upon commercially sourced PA-12 particles that can then be printed through regular LS approaches. We have demonstrated that the coated powders have the right extrinsic and intrinsic properties needed for LS and exploited these materials to print coloured and non-biofouling systems. The single-coloured powders can be physically mixed and printed, and final object colours can be predicted. Our methodology provides a working CMYK colour mixing system for LS printers and could open up a vast range of applications. Building upon this, we have shown that the same methodology can exploited to introduce anti-biofilm functionality and we have printed materials that demonstrably confer a significant reduction in biofilm formation across a range of morphologically and structurally diverse microorganisms.

Most importantly, we demonstrate a facile approach to introduce desirable functionalities onto polymer powder based commercial 3-D printing systems and we are confident that this same methodology could be exploited to deliver a wide range of other desirable functionalities and properties to printed objects.

## Methods

### Materials

Methyl methacrylate (MMA) (99 %) was obtained from Kaneka Belgium N.V. Isobornyl methacrylate (IBMA) (>80%) (CPDT) (97%) was purchased from Sigma Aldrich. 2,2′-Azobis(2-methylpropionitrile) (AIBN) was purchased from Fluorochem. CO$_2$ was purchased from Air Products. Triethylamine, petroleum ether, acetone, chloroform, hexane, and toluene were purchased from Fischersci. Dichloromethane (DCM), ethyl acetate, 4-phenylazophenol, (E)−2-(ethyl(4-((4-nitrophenyl)diazenyl)phenyl)amino)ethan-1-ol, and 1-(dimethylamino)−4-((2-hydroxyethyl) (methyl)amino)anthracene-9,10-dione, were purchased from Sigma Aldrich. Methacryloyl chloride was purchased from Acros Organics and Novozyme 435 was purchased from Novozymes. Disperse Red 1 methacrylate, Yellow 1 methacrylate, Disperse Blue 3 methacrylate were used as synthesised. PA-12 was purchased from EOS. PA-12 coated with P(IBMA-DB3MA) (Blue PA-12 ink), PA-12 coated with P(IBMA-DR1MA) (Red PA-12 ink), and PA-12 coated with P(IBMA-Y1MA) (Yellow PA-12 ink) were produced in house. Terpene acrylate/methacrylate and modified acid acrylate were used as synthesised.

### Methods

**Synthesis of disperse red 1 methacrylate (DR1MA).** To a cold solution (0 °C) of (E)−2-(ethyl(4-((4-nitrophenyl)diazenyl)phenyl)amino)ethan-1-ol (3.35 g, 0.0107 mol) in dichloromethane (83.75 mL) triethylamine (2.43 g, 0.024 mol) was added. To the resulting solution, methacryloyl chloride (1.569 g, 0.159 mol) was added dropwise, and left stirring at

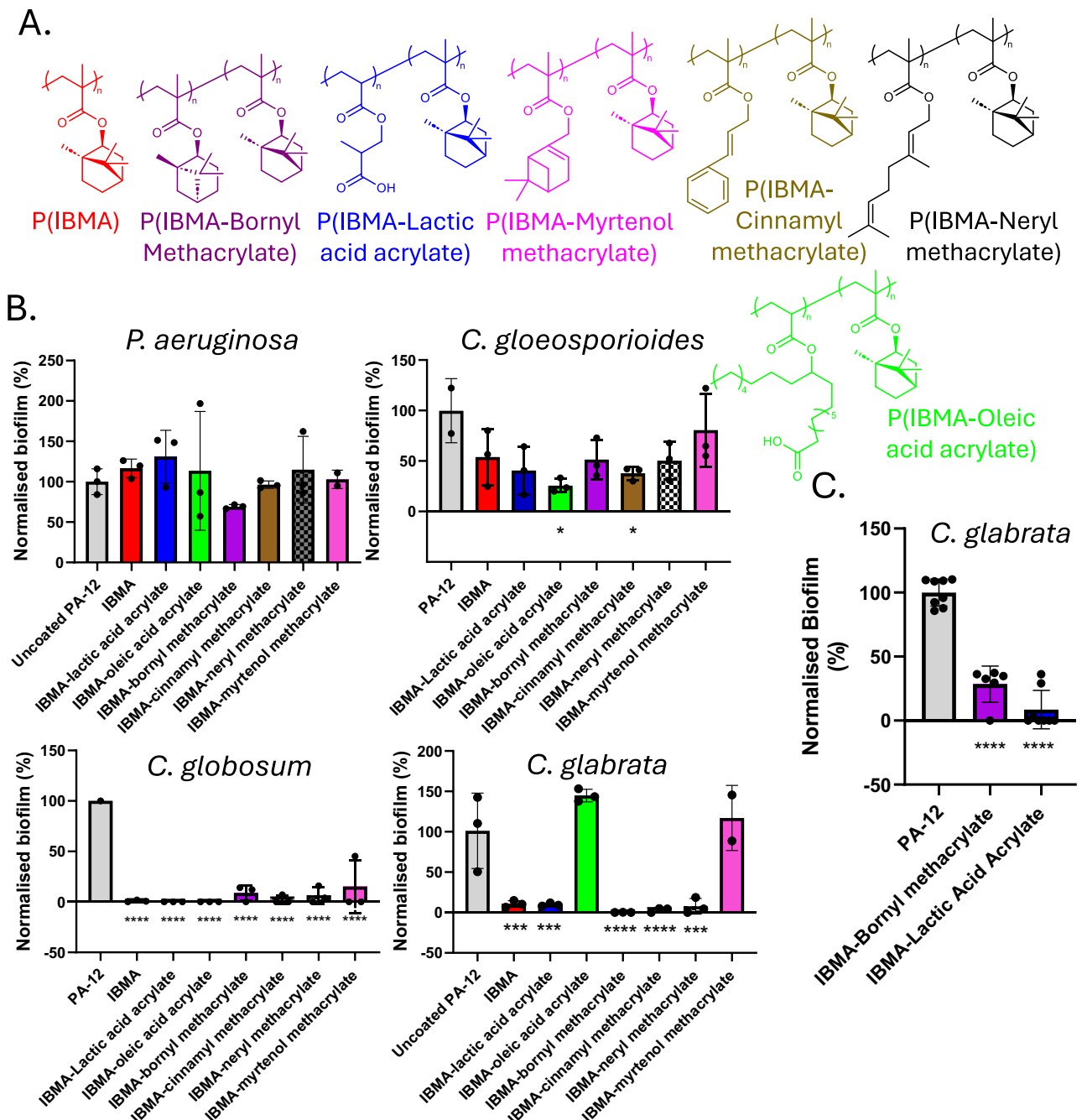

**Fig. 4 | Development of biofilm-resistant materials for SLS through scCO₂ polymerisation and coating process. A** Chemical structures of copolymeric coatings on PA-12 particles for anti-biofilm screening. **B** Biofilm assays on melt-pressed discs (5 mm × 1.5 mm) of PA-12 particles coated with p(IBMA-terpene monomer) or p(IBMA-modified acid acrylate). In all cases, biofilm activity on the surfaces was determined by XTT assay (Methods) and the absorbance expressed relative to that for biofilm on the uncoated PA-12, giving the normalised biofilm (%) for the indicated organisms. Colours of the data bars (left panel) correspond to the relevant chemical structures (top). Statistical significance was assessed by one-way ANOVA: *$p < 0.05$; **$p < 0.01$; ***$p < 0.001$; ****$p < 0.0001$. $n = 3$ biologically independent samples. The columns in the figure represent mean values; error bars represent standard deviation (SD). $p$-values (in parentheses) were significant for uncoated PA-12 versus: IBMA-Oleic acid acrylate (0.0145) and IBMA-cinnamyl methacrylate (0.0437) with *C. gloeosporioides*; every test polymer with *C. globosum* (<0.0001); IBMA (0.0002), IBMA-lactic acid acrylate (0.0002), IBMA-bornyl methacrylate (<0.0001), IBMA-cinnamyl methacrylate (<0.0001) and IBMA-neryl methacrylate (0.0001) with *C. glabrata*. **C** Biofilm assays for *C. glabrata* on SLS printed discs (5 mm × 1.5 mm) of PA-12 particles coated with p(IBMA-Bornyl methacrylate) and p(IBMA-Lactic acid acrylate). Statistical significance was assessed by one-way ANOVA: ****$p < 0.0001$. $n \geq 3$ biologically independent samples. The columns in the figure represent mean values; error bars represent standard deviation (SD). $p$-values for uncoated PA-12 versus IBMA and IBMA-lactic acid acrylate were <0.0001 in both cases.

---

500 rpm for 2 h. The reaction vessel was connected to a sodium hydroxide bubbler and sealed. The final mixture was quenched with saturated sodium carbonate solution (33 mL). The aqueous layers were extracted with dichloromethane, and the combined organic layers

were washed with brine, finally the organic solvent was evaporated under reduced pressure. The resulting red powder was dissolved in petroleum ether/ethyl acetate (40%/60%) and purified through a silica gel column adopting the same eluent mixture to yield a red powder

(after solvent evaporation) of disperse red 1 methacrylate. [1]H NMR (400 MHz, CDCl$_3$): δ 1.29 (3H, t, $^3J$ = 7.1 Hz, C$\boldsymbol{H}_3$), 1.98 (3H, s, C = CC$\boldsymbol{H}_3$), 3.57 (2H, q, $^3J$ = 7.1 Hz, NC$\boldsymbol{H}_2$CH$_3$), 3.76 (2H, t, $^3J$ = 6.2 Hz, NC$\boldsymbol{H}_2$CH$_2$O), 4.40 (2H, t, $^3J$ = 6.2 Hz, NCH$_2$C$\boldsymbol{H}_2$O), 5.62 (1H, s, C = C$\boldsymbol{H}$H), 6.13 ((1H, s, C = CH$\boldsymbol{H}$), 6.85 (2H, d, $^3J$ = 8.5 Hz, 2× Ar$\boldsymbol{H}$), 7.93 (2H, d, $^3J$ = 8.5 Hz, 2× Ar$\boldsymbol{H}$), 7.95 (2H, d, $^3J$ = 8.5 Hz, 2× Ar$\boldsymbol{H}$), 8.35 (2H, d, $^3J$ = 8.5 Hz, 2× Ar$\boldsymbol{H}$) ppm.

**Synthesis of Yellow 1 Methacrylate (Y1MA).** To a cold solution (0 °C) of (E)−4-phenyl azophenol (5 g, 0.025 mol) in dichloromethane (200 mL) triethylamine (5.78 g, 0.057 mol) was added. The reaction vessel was connected to a sodium hydroxide bubbler and sealed. To the solution, methacryloyl chloride (3.711 g, 0.355 mol) was added dropwise, and left stirring at 500 rpm for 4 h. The solution was quenched with saturated sodium carbonate solution (79 mL). The aqueous layers were extracted with dichloromethane, and the combined organic layers were washed with brine, and the solvent was evaporated. The resulting yellow-brown powder was dissolved in petroleum ether. The solution was purified through a silica gel column with petroleum ether as eluent, and the solvent was then evaporated. The product was then columned through basic alumina with petroleum ether, to yield a yellow powder of yellow 1 methacrylate. [1]H NMR (400 MHz, CDCl$_3$): δ 2.11 (3H, s, C = CC$\boldsymbol{H}_3$), 5.82 (1H, m, C = C$\boldsymbol{H}$H), 6.42 (1H, s, C = CH$\boldsymbol{H}$), 7.32 (2H, m, 2× Ar$\boldsymbol{H}$), 7.47–7.58 (3H, m, 3× Ar$\boldsymbol{H}$), 7.94 (2H, m, 2× Ar$\boldsymbol{H}$), 8.00 (2H, m, 2× Ar$\boldsymbol{H}$) ppm.

**Synthesis of disperse blue 3 methacrylate (DB3MA).** 1-(dimethylamino)−4-((2-hydroxyethyl) (methyl)amino)anthracene-9,10-dione (0.25 g, 0.0008 mol) was dissolved in toluene (5 mL); Novozyme 435 (0.5 g) and methyl methacrylate (1.02 g, 0.0119 mol) were added to the solution. The solution was raised to 55 °C and put under a vacuum of 0.075 MPa, and was left for 24 h. The solution was filtered, and washed with dichloromethane, the solvent was then evaporated. The resulting blue powder was purified through a silica gel column with an ethyl acetate eluent, yielding a blue powder of disperse blue 3 methacrylate. [1]H NMR (400 MHz, CDCl$_3$): δ 1.98 (3H, s, C = CC$\boldsymbol{H}_3$), 2.38 (3H, s, NC$\boldsymbol{H}_3$), 3.11–3.15 (4H, m, 2× CH$_2$), 5.710 (1H, s, C = C$\boldsymbol{H}$H), 6.25 (1H, s, C = C$\boldsymbol{H}$H), 7.28 (2H, s, 2× Ar$\boldsymbol{H}$), 7.72 (2H, m, 2× Ar$\boldsymbol{H}$), 8.36 (2H, m, 2× Ar$\boldsymbol{H}$), 10.64 (2H, s, 2× NH) ppm.

**Synthesis of PA-12 coated with P(IBMA-dye monomer) (60 mL autoclave).** PA-12 (10 g), IBMA (0.5 g, 0.0022 mol), Dye monomer (0.05 g, 0.18 mmol for Y1MA; 0.13 mmol for DR1MA; 0.14 mmol for DB3MA), and AIBN (0.025 g, 0.1522 mmol) were degassed in the autoclave for 15 min, by flushing it with CO$_2$ at a constant pressure between 0.27–0.41 MPa. The key was then shut rapidly, and the pressure was slowly increased to 5.51 MPa, subsequently the temperature was increased to 50 °C, and after stabilising the autoclave was pressurised to 13.79 MPa, with subsequent heating to 65 °C. Once stable at 65 °C the reaction is pressurised to 20.7 MPa and left stirring at 450 rpm for 24 h. The autoclave is allowed to cool <30 °C and the CO$_2$ is vented slowly, and the autoclave is opened to reveal the product. A variety of reaction conditions were used for these reactions.

**Synthesis of PA-12 coated with P(IBMA-dye monomer) (1 L autoclave).** PA-12 (120 g), IBMA (19.5 g, 0.0022 mol), Dye monomer (0.5 g, 1.87 mmol for Y1MA; 1.31 mmol for DR1MA; 1.37 mmol for DB3MA), and AIBN (0.2 g, 0.00122 mol) were degassed in the autoclave for 15 min, by flushing it with N$_2$ at a constant pressure between 0.27–0.41 MPa. The key was then shut rapidly, and the pressure was slowly increased to 5.51 MPa, subsequently the temperature was increased to 50 °C, and after stabilising the autoclave was pressurised to 13.79 MPa, with subsequent heating to 65 °C. Once stable at 65 °C the reaction is pressurised to 20.7 MPa and left stirring at 450 rpm for 24 h. The autoclave is allowed to cool <30 °C and the CO$_2$ is vented slowly, and the

autoclave is opened to reveal the product. A variety of reaction conditions were used for these reactions.

**Printing and fabrication of primary coloured parts**
Squares and tensile bars were designed (Magics, Design Software). The PA-12 powder coated with P(IBMA-Dye monomer) was sieved through a 200 μm mesh and dried for 1 h at 100 °C. The powders were sintered on an EOS Formiga P100, with varying conditions. The printed parts were left overnight and collected. The parts were cleaned and the powder was collected so it could be reused.

**Physical mixing of powders**
The calculated amounts of red and/or blue and/or yellow and/or white PA-12 based powders were added to a glass vial. The vial was physically mixed via a vortex mixer for 10 min, and subsequently left on a roller overnight, to ensure appropriate physical mixing.

**Printing of coloured structures**
The coloured powder mixtures were deposited onto the build platform of an EOS Formiga P100 SLS printer. A powder bed temperature of 177 °C, a chamber temperature of 150 °C, a hatching laser power of 20 W, a contour laser power of 20 W, a hatching speed of 2500 mm/s, a hatching distance of 0.25 mm, and a double scan were used for the printing of the coloured structures. Once cooled the structures were removed for visual inspection and colour analysis.

**Synthesis of PA-12 coated with p(IBMA-biologically active monomer)**
PA-12 (7.5 g), IBMA (1.1255 g, 0.005 mol), x monomer (10, 20, or 30 wt% compared to IBMA), and AIBN (0.0124 g, 0.0755 mmol or 0.0135 g, 0.0822 mmol or 0.0145 g, 0.0883 mmol) were degassed in the autoclave for 15 min, by flushing it with CO$_2$ at a constant pressure between 0.27–0.41 MPa. The key was then shut rapidly, and the pressure was slowly increased to 5.51 MPa, subsequently the temperature was increased to 50 °C, and after stabilising the autoclave was pressurised to 13.79 MPa, with subsequent heating to 65 °C. Once stable at 65 °C the reaction is pressurised to 20.7 MPa and left stirring at 450 rpm for 24 hours. The autoclave is allowed to cool <30 °C and the CO$_2$ is vented slowly, and the autoclave is opened to reveal the product.

**Multimaterial printing with mask strategy**
A mask of the object being printed is made and inserted into the powder, so that it can not move. The first powder is added to the printer, and the design is printed at the optimum conditions for the material. The first material is rapidly removed leaving the mask and the printed part in place, and the second material is added. The machine is warmed up to temperature, and the printing is continued. Once printing is completed the printer is left to cool and the part and mask are removed.

**Analysis**
**Nuclear magnetic resonance (NMR).** The samples were analysed by [1]H NMR, using a Bruker 400 Ultrashield (400 MHz NMR), and all the samples were dissolved in deuterated chloroform. [1]H NMR spectra were used to confirm formation of polymer and to calculate the monomer conversion.

**Gel permeation chromatography (GPC)**
GPC was performed in order to obtain the M$_n$, M$_w$, and dispersity (Đ) values for all the synthesised polymers. These values were produced using pMMA standards. All the measurements were performed on an Agilent 1260 Infinity HPLC using HPLC grade THF as eluent at 40 °C with 2× Agilent PL-gel mixed-D column at 1 mL min$^{-1}$ flow rate, connected to a differential refractive index (dRI) detector and MALLS detector.

## Scanning electron microscopy (SEM) and particle size analysis

SEM was used to see if the polymer materials were composed of discrete monodisperse particles, and it was performed using a Philips XL30 electron microscope. The samples were prepared by putting the polymer powder on a carbon tab mounted onto an SEM stub, which was then coated with Platinum for 180 s at 12 mA and 2.2 kV using an Emitech SC7640 sputter coater. The average size of the polymer particles was calculated by taking the area of 100 particles from a SEM image and then taking the average area of the particles exploiting ImageJ software.

## Laser diffraction spectrometry (LDS)

LDS was used to analyse the particle size of a large segment of the samples. This was done by using a Malvern Mastersizer 3000. Initially, the samples (1 g) were loaded onto a dispenser. Dispenser vibration moved the polymeric powder evenly into an air jet, which inserted the sample into the detector. Mastersizer 3000 software was then used to obtain the average particle size.

## Differential scanning calorimetry (DSC)

DSC was used to measure thermal transitions such glass transition temperature ($T_g$), temperature of crystallisation ($T_c$), and temperature of melting ($T_m$). The samples were prepared by weighing out circa 5 mg of sample into a DSC pan, which was then closed with a lid and tested from 0 °C to 250 °C. The $T_c$ was determined by finding the maximum point of the curve if any was present, the $T_m$ by finding the minimum point if any was present, and the $T_g$ by finding a change in the slope of the line if any was present.

## Colour of printed parts

The coloured parts were removed from the printer and placed onto a white sheet of paper. An image of the surface of the square was taken with a NIX colour sensor, and the data were compiled.

## Time-of-flight secondary ion mass spectrometry (ToF-SIMS)

ToF-SIMS data were acquired using a ToF-SIMS IV instrument (ION-TOF GmbH., Münster, Germany). The primary ion beam was a bismuth liquid metal ion gun (Bi3+) run at 25 kV (pulsed target current of -1 pA). For each sample, data were acquired over 3 regions of 500 μm × 500 μm at 256 × 256 pixels resolution for 20 scans. Data acquisition and analysis was performed using SurfaceLab 7 software (IONTOF GmbH). Selected spectra (area 1 of each sample) were normalised for comparison and identical Y scales were used.

## Biological assays

Biofilm metabolic activity was measured by the XTT (Sigma-Aldrich) reduction assay. First, non-infected discs were used as a negative XTT control, and these values were subtracted from the XTT absorbance obtained with the organisms. For *C. glabrata* BG2 and *P. aeruginosa* PAO1-L single colonies were used to inoculate, respectively, TSB broth or YPD broth cultures in Erlenmeyer flasks and incubated overnight at 37 °C with orbital shaking at 150 revolutions min$^{-1}$. Cultures were washed twice in RPMI 1640 and diluted to 125,000 cells ml$^{-1}$. Aliquots (100 μl) of the cell suspension were transferred to 96-well microtiter plates (Greiner Bio-One, Stonehouse, UK), either with melt pressed discs or 3D-printed discs composed of the desired polymers as described above and then incubated statically for 2 h. Similarly, 100 μl of spores ($2.5 \times 10^6$ spores ml$^{-1}$ in PDB) from 7-day-old PDA plates of the filamentous fungi *C. globosum* ATCC6205 or *C. gloeosporioides* ATCC38237 were transferred to coated 96-well plates for 6 h at room temperature.

In all cases, the discs were subsequently transferred to fresh 96-well plates. Non-adherent cells or spores were removed by three gentle washes with PBS; then, 100 μl of fresh medium was added to each well, and plates were incubated at 37 °C up to 24 h after inoculation. The melt pressed discs or 3D printed discs were again transferred to fresh plates. The wells were washed three times with PBS, and the XTT reaction was initiated by adding XTT and menadione to RPMI (for *C. glabrata* and *P. aeruginosa*) to final concentrations of 210 μg ml$^{-1}$ and 4.0 μM, respectively, or to PBS (for *C. globosum* and *C. gloeosporioides*) to final concentrations of 400 μg ml$^{-1}$ and 25 μM (final volume per well, 200 μl; PBS was used instead of PDB, as the XTT reaction does not work in PDB medium). After 2 and 6 h, respectively, 100 μl of the reaction solutions was transferred to fresh 96-well plates, and the absorbance at 490 nm was measured using a BioTek EL800 microplate spectrophotometer.

To assess potential deleterious impacts of the polymers on general (versus biofilm-only) fungal growth, i.e., toxicity, the procedure adopted was similar except that washing steps were omitted, so retaining non-biofilm cells, and the particles were not transferred to fresh plates, as described previously (Vallieres et al.). In the cases of *C. globosum* and *C. gloeosporioides*, as the XTT reaction cannot be performed in PDB medium, growth of these fungi with the polymers in PDB was assessed by $OD_{600}$ after 15 days.

## Mechanical properties analysis

The mechanical properties were investigated following ISO-527-2 standards (equivalent to ASTM D368) and were performed in an Instron Universal Testing system. Rate of extension chosen was 1.00 mm/min, humidity was 50%, and temperature was 18 °C.

## Water contact angle measurements

A Kruss DSA100 was used for water contact angle measurements of the surface of printed parts. Measurements were performed following the sessile drop method used for static contact angles.

## Surface roughness measurements

The area to be measured is a single field of view (2858 mm × 2176 mm) of two different locations in every specimen, one at the tab of the sample and another at the centre. Alicona G5 surface texture measurement has been identified as solution for these measurements (mode: 5X objective, vertical resolution of 1.50 mm, lateral resolution of 14.70 mm, z-vertical focus variation from −700 mm to +700 mm). Three repeats at the same location for every specimen under study will imply a total number of 54 surface measurements under the same conditions of temperature/humidity in the Manufacturing Metrology Team laboratory.

Data analysis was performed on Mountains software to analyse the surface texture measurements based on ISO 25178-2 and calculate height parameters of every measurement within the scale-limited surface:
(1)   Sq: root mean square height.
(2)   Ssk: skewness.
(3)   Sku: Kurtosis.
(4)   Sp: maximum peak height.
(5)   Sv: maximum pit height.
(6)   Sz: Maximum height (sum of maximum peak height and maximum pit height).
(7)   Sa: arithmetical mean height.

Refer to ISO25178-2 Terms, definitions, and surface texture parameters for further information.

## Vapour smoothing

Smoothing was performed via VaporFuse Surfacing (https://www.3dnatives.com/en/vaporfuse-surfacing-a-green-solution-for-improved-part-properties-03112020/#).

The samples were preheated for 15 min at 145 °C. Subsequently vapour was deposited utilising a minimum vapour pressure of

130 mbar for 11 s per cycle (20 cycles), after each cycle the sample was cooled to 110 °C.

## Reporting summary

Further information on research design is available in the Nature Portfolio Reporting Summary linked to this article.

## Data availability

Further data is available upon request to authors.

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

## Acknowledgements

The authors would like to thank the University of Nottingham Chemistry Workshops (Richard Wilson) for their help with high-pressure equipment and the Nanoscale and Microscale Research Centre (nmRC) staff for their help with imaging and processing. We would like to thank the LDMI (Low Dimensional Materials and Interfaces) doctoral training programme funding from the University of Nottingham. For PhD support (HRF), we also acknowledge the project code (EPSRC/SFI CDT in Sustainable Chemistry – Atoms 2 Products (EP/SO22236/1)).

## Author contributions

Eduards Krumins: The author planned, proposed, conducted experiments/analysis, and drafted/revised the manuscript. Liam Crawford: Co-author conducted and analysed biological testing. David M. Rogers: Co-author conducted and analysed computational chemistry in an effort to predict the colour of dye monomers. Fabricio Machado: Co-author conducted and analysed computational chemistry in an effort to predict the colour of SLS printed objects. Vincenzo Taresco: Co-author helped in the design of experiments and drafting of the manuscript. Mark East: Co-author provided technical assistance with AM work. Christopher Parmenter: Co-author conducted FIB-SEM imaging and processing. Samuel Irving: Co-author helped in the design of experiments. Harriet R. Fowler: Co-author aided in the experimental work regarding anti-biofouling coatings for PA-12 particles. Long Jiang: Co-author conducted and analysed ToF-SIMS experiments. Nichola Starr: Co-author conducted and analysed ToF-SIMS experiments. Kristoffer Kortsen: Co-author aided in the experimental work regarding anti-biofouling coatings for PA-12 particles. Valentina Cuzzocoli Crucitti: Co-author supported biological testing. Simon V. Avery: Co-author helped in the design of biological experiments and drafting of the manuscript. Christopher Tuck: Co-author was integral to the conception/design of experiments regarding AM and drafting/revision of the manuscript. Steven M. Howdle: Corresponding author was integral to the conception/design of experiments and in the drafting/revision of the manuscript.

## Competing interests

The authors declare no competing interests.
