## [Peer Review File · Nature Communications]

REVIEWER COMMENTS

Reviewer #1 (Remarks to the Author):

This manuscript reports a method for functionalizing polyamide powders suitable for SLS 3D printing technology, to achieve components with controllable and stable color and/or functional properties (i.e. bioactivity). The authors successfully achieved the proposed goals, in particular for what regards color of component, developing a method to calculate the point of color with the modified powders. This may have an important impact industry, while on scientific side seems limited. In fact, the literature in the field already showed similar approaches since several years to functionalize particles (e.g. doi: 10.1016/j.supflu.2018.01.030) or to obtain porous materials (e.g. doi : 10.1039/D1RA03341G). Furthermore, dye functionalization of polymers with scCO₂ was reviewed some years ago (doi: 10.1016/j.jcou.2021.101760). On the other hand, I agree that the work on Nylon is new, and this may have relevant impact, since nylon is the most used material in SLS.

The work is very solid, with well performed characterization, the results are consistent with the conclusions and reported with sufficient details, nevertheless they should be improved. For instance, fig 1A the X is not correct, and should be only the dye without the methacrylate part. Scale bar in figure 3d is too big and cover half of the picture, legend of x-axis in figure 4b is almost not readable.

Summarizing, in my opinion the work is very well performed, scientifically solid and with a huge support of tests. Nevertheless, in my feelings it can be more suitable for a more specialistic journal since the novelty is questionable and probably more relevant for industry purposes rather than scientific ones.

Reviewer #2 (Remarks to the Author):

The manuscript presents a method for introducing functionalities (e.g., multicolour and anti-biofouling) to polyamide-12 (PA-12) powders via supercritical polymerization, which makes an interesting progress on selective laser sintering (SLS) 3D printing. However, justification for conducting the supercritical polymerization is insufficient, especially for the colouring. In addition to spray painting or vat dipping, blending pigments or dyes is also a straightforward way, which can also resist scratching. I understand that the commercially available PA-12 powders are often white or grey in appearance. How the powders are prepared? If they can be coloured by simply blending pigments or dyes, the significance of the current work would be largely compromised. So, the necessity of the supercritical polymerization should be analyzed and discussed. For instance, pigments may be not able to be well dispersed in the powder particles, which is unfavourable for printing. Then, how about dyes? I recommend the manuscript to be published only when strong justification is presented.

Several additional issues which should be addressed are listed below.

1. In contrast to the solid and sound contents concerning the synthesis, the material parts are weak. Mechanical properties should be characterized via for instance tensile tests. The influence of coating various polymers on the mechanical strength of the printed objects should be studied.
2. More printed objects with diversified complex geometries should be presented to show the model-insensitive character of SLS printing. The structure of Master is not so challenging.
3. Resolution of the 3D printed objects should be studied quantitatively. It seems that neither the finishing surface nor the details of the printed Master is good.

Reviewer #3 (Remarks to the Author):

In this manuscript, the authors present a novel approach to surface coating particles for laser sintering (LS) to achieve goals of coloring or biofilm prevention. This technique expands upon previously described LS technology, specifically integrating industry-standard components and readily available materials for immediate integration and impact. The authors present a thorough analysis of coating efficiency, identification, color prediction, and coating effectiveness throughout the paper and supplemental information. However, several improvements/refinements should be made before carrying this article to publication to address deficiencies in data presentation, substantiation of claims, and repeatability as outlined below:

(a) One of the largest immediate items noted by the reader is the lack of printed examples. It is unclear which images are computer rendered versus printed, and in the presumed printed examples, there are no scale bars or size comparisons. To the reader, the only immediately recognized printed parts are Figure 2C (science fiction character) and Figure 3C (building block). It is unclear if Figure 2A, Figure 2D, Figure 3A, and Figure 3D is a printed part or not. The backgrounds of each of these parts are photoshopped to a white background, which, while this looks cleaner, makes it difficult to differentiate performance versus rendering. It is furthermore unclear what size or resolution is achieved in testing the product given lack of scale bars, demonstration prints, etc. We recommend integrating benchmark 3D models to demonstrate this aspect and introducing scale bars or scaled features (bench, lattice, cube, etc.).

(b) An argument is presented as to if coating affects polymeric properties including T_g , T_m , and T_c . However, it is broadly assumed throughout the paper that the materials properties of the final part are unchanged if the coating does not significantly change polymer material properties. This claim is unsubstantiated in the text and SI. A discussion in the SI is warranted as to how and why T_g , T_m , and T_c of the LS particles would or would not change with surface coatings introduced in this work. Dog bones should be printed and standard exams be carried out with uncoated particles as a control.

(c) There is a great discussion as to an initial computational model to prevent absorbance shifts upon methacrylation. This is further discussed in the SI section. However, there is no presentation as to the computationally determined spectrum of potential and final choices or as to how the peaks shift, increase, decrease, etc. that would be required to substantiate the computational choice. Key peaks are discussed in the SI, however, perceived color is not defined by few peak maxes in an absorbance spectrum and rather full visible spectrum shifts.

(d) There is a brief point in the paper where it is stated that shell co-polymers are not chemically bound, leading to an inconclusive finding of surface functionalization correlation to intended incorporation and subsequent conclusions made from this point. Please elaborate on what is meant by this statement and subsequent impacts.

(e) Several claims are made about the SEMs of mass amounts of particles (up to 100), however, these images are not shown in the paper or SI.

(e.1) A discussion is warranted as to why the surfaces shown in Figure 1C and Figure 2B look fibrous upon coating, what potential differences in coating efficiency are exhibited between dyes, and/or heterogeneity of coatings or coating method.

(f) Figure 3D has a large scale bar and image covering most of the SEM image, it is difficult to discern texture or macromolecular features because of this. It would be great to see a zoomed out cross section of this part to see differences in face surface texture.

(g) Interestingly, in the discussion of achieving a goal color set, there is a very in-depth exploration into achieving the intended color, but it is unclear as to why previous approaches for similarly resolved issues were inaccurate. For example, this issue has been addressed previously in inkjet printing, where it is well understood how to mix a set of colors together to yield a perceived color (given how human eyes have bias in interpreting different colors to different intensities). It may be advantageous to expand upon this further (perceived versus intended color) as this is potentially a large outcome of this work. Why not mix in black/white particles as well to achieve a color similar to the requirement for Key in inkjet printing?

(h) There are several locations where sentences are cutoff or missing dependent clauses, leading to difficulties in understanding the science conveyed.

(i) Figure 4B is inconsistent in formatting and display of error bars.

(j) The discussion of anti-biofilm assays and SEMs needs further substantiation.

(j.1) The XTT assay compares reaction supernatant of each printed disk against a non-coated disk, however, there is no negative control for each printed disk not exposed to bacteria or fungi, so it is unclear how or if the XTT assay was affected by different coatings leading to indeterminate differences in assay conditions. For instance, leaching of a component (as potentially mentioned earlier in the text) may lead to absorbance or efficiency differences in the assay. Therefore, the conclusion that lack of biofilm formation is attributed solely to functionalization is unsubstantiated.

(j.2) The SEMs of coated/uncoated components, specifically Figure 4D, are unclear and do not provide much, if any, additional substantiation. There are alternative methods for imaging biofilm coatings on surfaces that would provide greater evidence for lack of a visual coating or lack thereof.

(j.2.1) In resolution provided, it is difficult for the reviewer to make any further observations of surface irregularities given the SEMs are unfortunately provided blurry.

(j.3) While the statistical test is well described, there is no discussion of what the error bars represent within the paper.

(k) There is overall a lacking discussion as to the final properties of each surface coating including mentioned surface topology, roughness, hydrophobicity, and charge as well as fundamental mechanical properties expected for materials characterization. Including these findings would greatly strengthen this paper and broaden the field applicability.

Supplemental Information:

(L) Please see above for some points made as to additions required to the supplemental section.

(m) In all ¹H-NMRs shown, there are several items that need to be addressed: ordering of labelled peaks, labelling of peaks on the given molecule, labelling of unintended residuals (for instance, water), presentation of integrations values and splitting, and discussion of any expected or unexpected impurities. In several of these NMRs, peaks are present that should be attributed to the molecule but are not, hydrogens that should be present are unlabeled or not visible in the spectra, etc. – this makes it difficult to substantiate that the intended molecule was synthesized and purified.

(o) Several axes' labels are missing. If there are multiple dependent axes in one graph, please include and label all, otherwise it is unclear what is being measured, for instance in S9.

(p) Claims of DRI and UV peak overlap should be substantiated with peak max and FWHM or similar parameters to substantiate non-statistically significant differences in peak occurrence.

(p.1) This same claim goes for comparing particle size. This would be much clearer if presented in a singular graph and statistical conclusions made about how much larger the particle is after each coating and statistical significance.

(q) S22 and similar graphs are hard to interpret at the current formatting. Independent and dependent axes should be blown up to better show the intended data (for instance, for S22, setting independent range to max 25 minutes and dependent range -1 to 1 (units?)).

There exists work in similar spaces as outlined in the following; however, the combination of scCO₂, color ramifications, and approach are sufficiently novel. The closest applicable work relies on inorganic components and does not address color whereas this work focuses on organic components leading to color properties (same research group).[REF 1] Coupling use of scCO₂ in pre/post-preparing materials for LS is also not fully novel as other articles reference use of scCO₂, particularly in foaming applications.[REF 2] Other post-processing steps for surface functionalization have been previously described, but do not encompass this work.[REF 3] As referenced in the work, functionalization to achieve anti-biofilm surfaces has previously been extensively studied,[REFS 4,5] but not in combination with the additional features and goals incorporated in this work. Combination of colored components as computationally modelled in this work has been extensively described previously in inkjet printing related patents and literature; the paper herein mentions and gives credit to base models for producing these results.

References:

1. R. Larder, R. et al. Antimicrobial 'inks' for 3D printing: block copolymer-silver nanoparticle composites synthesised using supercritical CO₂. *Polymer Chemistry* 13, 3768–3779 (2022).
2. Yang, C., Chen, N., Liu, X., Wang, Q. & Zhang, C. Coupling selective laser sintering and supercritical CO₂ foaming for 3D printed porous polyvinylidene fluoride with improved piezoelectric performance. *RSC Advances* 11, 20662–20669 (2021).
3. Dizon, J. R. C., Gache, C. C. L., Cascolan, H. M. S., Cancino, L. T. & Advincula, R. C. Post-Processing of 3D-Printed Polymers. *Technologies* 9, 61 (2021).
4. Vallieres, C. et al. Discovery of (meth)acrylate polymers that resist colonization by fungi associated with pathogenesis and biodeterioration. *Science Advances* 6, eaba6574 (2020).

5. Sun, X. et al. Antibacterial Adhesion of Poly(methyl methacrylate) Modified by Borneol Acrylate. ACS Appl. Mater. Interfaces 8, 28522–28528 (2016)

Addressing reviewer comments:

Thank you for the opportunity to respond to the reviewer comments. We have taken these very seriously and some of the comments have definitely helped us to strengthen the quality of our manuscript and to clarify the story.

We have used a colour code in order to help. The review comments are in **bold**. Text that is in **red** indicated changes to the main manuscript. Text that is in **yellow** indicates changes to the supporting information.

Reviewer One:

Comment 1: In fact, the literature in the field already showed similar approaches since several years to functionalize particles (e.g. doi: 10.1016/j.supflu.2018.01.030) or to obtain porous materials (e.g. doi : 10.1039/D1RA03341G). Furthermore, dye functionalization of polymers with scCO₂ was reviewed some years ago (doi: 10.1016/j.jcou.2021.101760).

The reviewer has misunderstood the main thrust of our new approach. We have taken commercially available PA-12 particles, the most common polymer used commercially in SLS. We have used scCO₂ to colour those particles very efficiently. We then print coloured 3-D objects with absolutely conventional SLS apparatus.

The reviewer highlights other papers where scCO₂ has been exploited in very different ways to carry out post processing of printed objects.

Paper one: doi: 10.1016/j.supflu.2018.01.030 "Drug impregnation for laser sintered poly(methyl methacrylate) biocomposites using supercritical carbon dioxide"

- *The application is related to drug release.*
- *Articles have been printed from PMMA particles. This 3-D object is then impregnated with a drug. Therefore, this is a post-printing procedure utilizing scCO₂.*

Paper two: doi: 10.1039/D1RA03341G "Coupling selective laser sintering and supercritical CO₂ foaming for 3D printed porous polyvinylidene fluoride with improved piezoelectric performance"

- *Again, this application is clearly different to ours taking a 3-D printed PVDF object and treating it with scCO₂ to create a porous/foamed PVDF printed part. So, like Paper 1, this is a post-printing treatment.*

Paper three: doi: 10.1016/j.jcou.2021.101760 "Supercritical CO₂-assisted dyeing and functionalization of polymeric materials: A review of recent advances (2015–2020)"

- *This review explains how scCO₂ can be used for impregnation of polymeric materials with dyes/insecticides/drugs – e.g. postprocessing of a 3-D printed part.*

Based upon the above points, there are no changes to the manuscript.

Comment 2: Fig 1A the X is not correct, and should be only the dye without the methacrylate part.

The referee was correct. This was our error and we have corrected Figure 1 (methacrylate group removed from Figure1A-X) as recommended by the reviewer.

Comment 3: Scale bar in figure 3d is too big and cover half of the picture.

Figure 3 has been changed based on the recommendation of the reviewer. (The scale bar on SEM image has been reduced in size, and a larger area of the image has been shown).

Specifically see 'D' in figure 3.

Comment 4: Legend of x-axis in figure 4b is almost not readable.

Figure 4 has been changed based on the recommendation of reviewer 1 and 3. The graphs have been made larger, so that the x-axis labels are more reader friendly, SEM images have been removed, and error bars have been corrected:

- *This can be seen in the reproduced figure below:*

A.

B.

C.

Reviewer 2:

Comment One: Why use scCO₂ instead of another procedure: Vat dyeing, spray dyeing, blending pigments, blending dyes, other coating/polymerization techniques?

The reviewer clearly is asking "why bother to use scCO₂ when there are lots of other methods for adding colour". The reason is that conventional approaches simply cannot deliver the modified particles that are required to print in colour. Up until now the main approach had been to paint the surfaces of the 3-D objects after they have been printed (from white particles) either by vat dyeing or spray dyeing, but as we explained (see page 3 of the originally submitted manuscript) this causes problems with reproducibility and the materials might be scratched.

A better route would be to print in colour but this requires coloured particles and they do not exist. PA-12 is the ubiquitous polymer for SLS and is used worldwide. We have focused on this material and our approach has been to coat each PA-12 particle with a thin layer of a coloured polymer such that the new coloured particles can be easily printed in commercial apparatus. Blending dyes into PA-12 particles is not possible. Dispersion, precipitation and suspension polymerisation approaches in conventional solvents do not give uniform coverage across the PA-12 substrate and they each require the addition of extra materials such as surfactants. The scCO₂ method we describe gives perfect coverage across each particle (as shown by the SEM data) and does not require additional reagents such as surfactants or stabilizers. The scCO₂ approach is easily scalable and is very efficient; ~500 g of material can be produced in a 1 L vessel, and the CO₂ can be recycled and reused. Additionally, this is a sustainable method for isolating the dry powder, as energy intensive solvent drying is not needed.

Most importantly, as emphasized in the manuscript, the CO₂ process works across a wide range of differing polymeric cores and shell materials, hence it has the potential to be a platform technique that can be utilized to produce a variety of different functional materials.

There are no changes to the manuscript.

Comment two: How are PA-12 particles made?

The PA-12 particles that we have used are commercially obtained as is stated in the materials section of the SI. To answer the reviewer's question though, the particles can be made through a variety of methods.

The SLS process requires high-quality powders with precise control over their size and shape. Commercial powders for SLS typically consist of a majority of particles around 60 µm in size, along with a smaller fraction having an average size below 10 µm.

There are various methods for producing powders for SLS, with precipitation and mechanical grinding being the most important ones. The choice of method depends on the material and desired powder properties. In some cases, powder preparation is just one step in a multi-step fabrication process.

For instance, PA-12 can be produced through hydrolytic or anionic ring-opening polymerization of lauryl lactam in organic solvents. The precipitation process yields PA-12 powders with controlled size distribution and advantageous thermal properties for SLS processing.

Spherical particles for SLS can be obtained through methods like coextrusion, emulsion polymerization, or anionic ring-opening precipitation polymerization with nucleating agents.

Alternatively, powders can be produced by mechanical grinding, particularly cryogenic milling, which can provide particles smaller than 100 μm . However, these powders may have lower flowability and can result in mechanically weaker parts.

Additionally, cryogenic grinding can be used to blend different polymers during the process, allowing for the creation of unique SLS powders. This blending process involves particle fracture, flow, and welding induced by high-energy ball-powder-collisions in a vibratory ball mill.

In summary, SLS powders require careful control of size and morphology, and their production methods vary depending on the material and desired properties, including precipitation, coextrusion, emulsion polymerization, cryogenic milling, and mechanical blending.

These approaches have been reviewed extensively; Ligon, S. C., Liska, R., Stampfl, J., Gurr, M. & Mülhaupt, R. *Polymers for 3D Printing and Customized Additive Manufacturing*. *Chem. Rev.* **117**, 10212–10290 (2017).

There are no changes to the manuscript.

Comment Three: In contrast to the solid and sound contents concerning the synthesis, the material parts are weak. Mechanical properties should be characterized via for instance tensile tests. The influence of coating various polymers on the mechanical strength of the printed objects should be studied.

This was an excellent observation by the reviewer and one that caused us to go back and measure very carefully the materials we have made. Mechanical testing was performed on the printed coloured materials and compared directly to a control of commercial PA-12 printed via the same method. The "take home" message is that the mechanical properties of printed PA-12 are actually improved by adding in coloured coating in the 80:20 cases. We found an increased elastic region and higher yield stress in the 80:20 materials compared to the 100% commercial PA12 control materials .

To the manuscript text the following was added: Coloured and functional PA-12 based materials can be processed easily by SLS to produce 3-D objects with very good mechanical properties. In fact the 80/20 % mix of commercial PA-12 and PA-12 coated with P(IBMA-DR1MA) showed an increased elastic region and higher yield stress compared to the materials printed under identical conditions using the commercial PA-12 materials (S59).

To the SI the following was added: Mechanical testing was conducted on tensile bars printed using ISO-527-2 standards. These standards are known to be equivalent to ASTM D638 standards. Both are the international standard for tensile testing of rigid and semi rigid thermoplastic molded, extruded, and cast materials. Four differing compositions were chosen:

- 100% virgin PA-12 (control)*
- 100% PA-12 coated with P(IBMA-DR1MA) (Red)*
- 50/50 % mix of two powders; PA-12 coated with P(IBMA-DR1MA) and virgin PA-12*
- 80/20 % mix of two powders; virgin PA-12 and PA-12 coated with P(IBMA-DR1MA) respectively*

The only difference in printing conditions of the tensile bars was heating rate, as the powders containing PA-12 coated with P(IBMA-DR1MA) necessitated a slower heating rate (1.5 min/degree compared to 1 min/degree) to prevent curling. Having the same sintering conditions allows for reliable mechanical property comparisons of the printed objects. All of the tensile bars were printed flat with the long axis in X orientation (S56D). Testing revealed that mechanical properties are altered through the addition of the functional coatings (S59).

S59 – Mechanical properties testing (Stress vs Strain) using ISO 527-2 standard tensile bars composed of four differing compositions, 100% virgin PA-12 (control; blue group) and three coloured samples built from 100% PA-12 coated with P(IBMA-DR1MA)(dark red group), a 50/50 % mix of PA-12 coated with P(IBMA-DR1MA) and virgin PA-12 (green group), and an 80/20 % mix of virgin PA-12 and PA-12 coated with P(IBMA-DR1MA) respectively (yellow group).

It is evident that when utilizing an 80/20 mix of virgin PA-12 and PA-12 coated with P(IBMA-DR1MA) that the coloured tensile bars (S59-yellow group) have an increased elastic region and higher yield stress than those of the control materials (S59 - blue group). This could be because of increased bonding or possibly higher part density. This could potentially be due to higher absorptivity of the coloured particles coupled with the standard material. This effect is reversed with increasing strain % of the coated materials, meaning further optimisation might be necessary (S59). The results indicate that our coloured materials would be commercially viable as they have very similar mechanical properties to virgin PA-12. An important fact to note is that the linear segments for all the tested tensile bars are quite similar, suggesting that they behave similarly before the ultimate tensile strength is reached.

Comment Four: More printed objects with diversified complex geometries should be presented to show the model-insensitive character of SLS printing. The structure of Master is not so challenging.

Additionally, Reviewer 3 made a similar comment

Reviewer 3 Comment A: One of the largest immediate items noted by the reader is the lack of printed examples. It is unclear which images are computer rendered versus printed, and in the presumed printed examples, there are no scale bars or size comparisons. To the reader, the only immediately recognized printed parts are Figure 2C (science fiction character) and Figure 3C (building block). It is unclear if Figure 2A, Figure 2D, Figure 3A, and Figure 3D is a printed part or not. The backgrounds of each of these parts are photoshopped to a white background, which, while this looks cleaner, makes it difficult to differentiate performance versus rendering. It is furthermore unclear what size or resolution is achieved in testing the product given lack of scale bars, demonstration prints, etc. We recommend integrating benchmark 3D models to demonstrate this aspect and introducing scale bars or scaled features (bench, lattice, cube, etc.).

Comment Five: Resolution of the 3D printed objects should be studied quantitatively. It seems that neither the finishing surface nor the details of the printed Master is good.

We have taken these comments on board and we accepted the challenge!! We have added to the manuscript a much wider range of printed materials with varying scale and geometry that have been printed with our new and versatile coloured PA-12 based materials. Amongst these are a challenging lattice type structure (recommended by reviewer 3) and an industry standard test piece that demonstrates the resolution that is obtained in SLS.

To the text the following was added: *Other 3-D objects with a range of geometries and scales were also printed using these new coloured PA-12 materials. Detailed analysis revealed that resolutions of ~100 µm could be achieved and dimensional error when comparing design to finished part was less than 0.5% (S54).*

To the SI the following was added: *A variety of further parts were printed with both the PA-12 coated with P(IBMA-DB3MA) (blue) and P(IBMA-DR1MA) (red) in varying ratios and on a range of scales and geometries. A design for testing resolution of printed parts was built from a 80/20 mix of virgin PA-12 and PA-12 coated with P(IBMA-DB3MA)(S54).*

S54 – A.) CAD model of an industry standard printed resolution test model B.) Printed model built from a a 80/20 mix of virgin PA-12 and PA-12 coated with P(IBMA-DB3MA) respectively. C.) Table of actual measurements of the printed part vs the design targets. The printed parts show a resolution of ~100 µm and an overall error in dimension of less than 0.5%.

A model race car which has interlocking parts (wheels interlock with the main body of the car) was constructed from a 80/20 mix of virgin PA-12 and PA-12 coated with P(IBMA-DB3MA) respectively (S55).

S55 – Printed race car model built from a 80/20 mix of virgin PA-12 and PA-12 coated with P(IBMA-DB3MA) respectively. A.) fully assembled. B.) with detachable wheels removed.

Tensile bars were printed from PA-12 coated with P(IBMA-DR1MA) (red) and a series of blends (S56).

S56 – Tensile bars A.) built from PA-12 coated with P(IBMA-DR1MA). B.) from a 50/50

mix of PA-12 coated with P(IBMA-DR1MA) and virgin PA-12. C.) from an 80/20 mix of virgin PA-12 and PA-12 coated with P(IBMA-DR1MA) respectively. D.) CAD model of the tensile bars showing build orientation.

A lattice structure was built from PA-12 coated with P(IBMA-DR1MA), showing that a complex intertwined part can be built with the coated PA-12 based material (S57).

S57 – Images of lattice type structure built from PA-12 coated with P(IBMA-DR1MA).

For the surface finish, the referees comments have led us to undertake some further investigations and process our printed materials with vapour smoothing, which is an industry standard technique. Previously, all samples were only treated by sand blasting, a rudimentary post-printing process. After sandblasting our coloured samples and those built from commercial PA-12 have very similar surface finishes. In order to increase surface smoothness the vapour smoothing step increases the surface finish significantly (S58).

To the SI the following was added: An industry standard surface finish can be achieved following vapour smoothing (commonly used post-processing tool). Vapour smoothing was used on a toy block printed with and 80/20 mix of virgin PA-12 and PA-12 coated with P(IBMA-DB3MA) respectively (S58).

S58 – Images of non-smoothed and post vapour smoothed printed parts showing an improvement in surface finish.

Reviewer 3:

Comment B: An argument is presented as to if coating affects polymeric properties including T_g , T_m , and T_c . However, it is broadly assumed throughout the paper that the materials properties of the final part are unchanged if the coating does not significantly change polymer material properties. This claim is unsubstantiated in the text and SI. A discussion in the SI is warranted as to how and why T_g , T_m , and T_c of the LS particles would or would not change with surface coatings introduced in this work. Dog bones should be printed and standard exams be carried out with uncoated particles as a control.

We have addressed this comment by providing a more through explanation of the design of our materials and further mechanical testing. See response comment 3 of reviewer 2.

The thermal properties of the coated vs uncoated particles are very similar, because the thermal properties of the coating and the core of the particle are similar. The coated particles were designed in such a way that the core (PA-12) has a T_m that is matched by the T_g of the coating we add (predominantly PIBMA). The T_m of PA-12 can range between 170-180°C and the T_g of PIBMA ranges from 160-180°C. The range of temperatures for PIBMA is larger than that of PA-12, however when using polymers above a certain molecular weight (>30,000) the T_g of PIBMA is between 170-180°C. In the observed materials the molecular weights are above this threshold, hence the T_g of the coating and T_m of the core do definitely match, leading the uniform sintering. The T_g of PIBMA ensures that the coating and core will sinter simultaneously leading to good printing. It should be noted that the T_g of PIBMA is significantly higher than the T_c of PA-12, hence no problems such as curling or warping are seen.

The coatings we use are of course copolymers of P(IBMA-Dye monomer) or P(IBMA-Bioactive monomer), in both cases we should stress that the vast majority of the copolymer is comprised of PIBMA, hence the thermal properties of the copolymers are analogous to those of PIBMA. Therefore, the functional coatings are compatible with the PA-12 core. This strategy could be utilized to coat a variety of other polymers as long as the core and the coating are designed in such a way that the thermal properties of both

are compatible.

No changes to the manuscript will be made other than those written in the answer to comment 3 of reviewer 2.

Comment C: There is a great discussion as to an initial computational model to prevent absorbance shifts upon methacrylation. This is further discussed in the SI section. However, there is no presentation as to the computationally determined spectrum of potential and final choices or as to how the peaks shift, increase, decrease, etc. that would be required to substantiate the computational choice. Key peaks are discussed in the SI, however, perceived color is not defined by few peak maxes in an absorbance spectrum and rather full visible spectrum shifts.

Predicting the exact position of the UV maxima is difficult, but what we did here is to use computational approaches to demonstrate that the addition of a methacrylate group should cause minimal shift in wavelength. For the dyes that we chose, the computational approach predicted a shift of 8 nm i.e. a minimal change to the spectrum and the colour of the material. This is what we then observed when we made the dye monomers.

The following text was added: Computed UV spectra for the azobenzene and anthraquinone hydroxy and methacrylate monomers show (S6) that for azobenzene there is a predicted blue shift of 8 nm for the second vertical transition upon changing from hydroxy to methacrylate. For anthraquinone the blue shift for the first vertical transition is predicted to be less than 1 nm upon changing from hydroxy to methacrylate. These blue shifts are considered negligible.

S6 - Computed UV spectra for azobenzene hydroxy (red line) and azobenzene methacrylate (green line, upper panel), and anthraquinone hydroxy (red line) and anthraquinone methacrylate (green line, lower panel). Gaussian functions of half width 12.0 nm were fitted to the TDDFT calculated vertical transitions for the gas-phase optimised geometries in solvent (DCM).

Comment D: There is a brief point in the paper where it is stated that shell copolymers are not chemically bound, leading to an inconclusive finding of surface functionalization correlation to intended incorporation and subsequent conclusions made from this point. Please elaborate on what is meant by this statement and subsequent impacts.

We thank the referee. We have rewritten a section of text in the document to better explain our description: The coloured copolymer forms a physical coating during the polymerization in scCO₂. The surfaces of the PA-12 particles act as a locus for the growing coloured copolymer chain and as this precipitates from the continuous phase scCO₂ it forms on the surface. But should be noted that the amorphous regions of PA-12 are significantly swollen by the scCO₂ and as such the PA12- surface is penetrated by the growing polymer chain. Thus, leading to interpenetration at the surface layers. The coating can be removed by selected solvents in which the copolymeric coating is soluble, such as chloroform or tetrahydrofuran. However, the copolymers are insoluble in water and leaching tests have been undertaken over a period of 6 months showing that in aqueous environments, the coating does not leach at all. In addition, it must be pointed out that through the sintering process there is significant enhancement of the interpenetration via melting, and in the final printed parts we see no evidence of any partition or separation at the resolutions we can interrogate.

Comment E: Several claims are made about the SEMs of mass amounts of particles (up to 100), however, these images are not shown in the paper or SI. (e.1) A discussion is warranted as to why the surfaces shown in Figure 1C and Figure 2B look fibrous upon coating, what potential differences in coating efficiency are exhibited between dyes, and/or heterogeneity of coatings or coating method.

We thank the reviewer for the comments about the SEM images, we have taken these into account and added the below SEM images and text to the SI as follows:

S14 – SEM image of PA-12 particles coated with P(IBMA-Y1MA)(Yellow).

S20 – SEM image of PA-12 particles coated with P(IBMA-DR1MA)(Red).

S24 – SEM image of PA-12 particles coated with P(IBMA-DB3MA)(Blue).

S25 - SEM of commercial virgin PA-12 (uncoated) showing the characteristic 'potato' shape and the range of particles sizes that are present in the commercial sample.

The SEM images (S14, 20, 24, and 25) demonstrate that the commercial PA-12 particles are not significantly changed by our coating process; the SEMs look the same before and after coating. There are smaller particles present initially from the commercial process used to make the (PA-12) and these are not changed significantly after our colour coating process.

When we focus right in at high resolution (Figure 1, 2) it is possible to see that the coating on the surface does look different, there is a slightly fibrous look to the materials and this definitively proves that we have neatly coated all the particles very effectively. In extensive analysis of the SEMs we found no "rogue" particles that were not coated or were formed of just the dye copolymer alone. This shows that the scCO₂ polymerisation is incredibly clean and effective as well as being solvent and surfactant free.

We did not see any differences in the behaviour of the three different copolymer dyes; all were found to coat the PA-12 effectively. The coating efficiency will be very similar because the monomers are all similarly soluble in scCO₂ at the reaction conditions and their methacrylate nature and the size of the dye functionality means that the polymerisation reaction rates will be similar. Also, as the polymers form they will quickly reach a chain length that is insoluble in the scCO₂ phase and they will all similarly precipitate onto the PA-12 particles surfaces as described in the manuscript. Why similarly? Because the bulk of the copolymer is the IBMA monomer in each case. Differing loadings of the dye monomers in the copolymers are used not because of coating efficiency, but because some of the dye monomers have a more intense colour than others.

Comment F: Figure 3D has a large scale bar and image covering most of the SEM image, it is difficult to discern texture or macromolecular features because of this. It would be great to see a zoomed out cross section of this part to see differences in face surface texture.

Figure 3 has been revised, see reviewer 1.

Comment G: Interestingly, in the discussion of achieving a goal color set, there is a very in-depth exploration into achieving the intended color, but it is unclear as to why previous approaches for similarly resolved issues were inaccurate. For example, this issue has been addressed previously in inkjet printing, where it is well understood how to mix a set of colors together to yield a perceived color (given how human eyes have bias in interpreting different colors to different intensities). It may be advantageous to expand upon this further (perceived versus intended color) as this is potentially a large outcome of this work. Why not mix in black/white particles as well to achieve a color similar to the requirement for Key in inkjet printing?

We thank the reviewer for the opportunity to further clarify our process. We agree that in the inkjet approach this has been addressed by others. But there has never been a colour mixing approach in SLS and this is very different in the solid phase. We have adapted and learnt from the inkjet approach but we have also needed to try out some very new approaches.

We have made changes to both the manuscript and the SI to clarify this point.

To the text the following was added: It is well known that predictive algorithms and other methods have been used to accurately and reliably predict the colour of formulations used in coloured inkjet printing. However, there are added complexities in a colour mixing system for SLS such as the possibility of colour change during melting, which necessitate the need for novel predictive methods (S38).

To SI the following was added: Our strategy for colour mixing was to emulate the Cyan, Magenta, Yellow, and Black (CMYK) colour mixing that is prevalent in ink jet paper printers. This is because even though the mixed powders are a solid, when sintered the melt pool is a liquid so the closest analogue for colour mixing is CMYK or subtractive colour mixing.

There are known approaches for predicting CMYK colour mixing used in inkjet printers. This has been achieved through a variety of methods such as the compilation of millions of data points recording data of individual formulations or computational methods such as the use of genetic algorithms.

The key reason why we can not use the established methods for the colour mixing exhibited in this work is that the colours can change significantly during the melting and sintering. We found that this results in the colour of a printed red part being different than the colour of the polymeric powder used. This can be clearly seen in the figure below (S38)

S38 –SLS Printing of tensile bars showing colour change on sintering. A. The top layer of powder as it has been spread in this case the 80/20 mix of virgin PA-12 and PA-12 coated with P(IBMA-DR1MA) respectively B. Tensile bars being printed. Note the distinct darkening in the sintered material.

The differences in colour between the powder and the molten part add complexity and this is why novel methods for colour prediction were developed. We utilized a combination of particle swarm optimization and multivariable analysis to predict this change and accurately predict the colours of the sintered formulations.

In all of the samples we have printed we have added significant levels of virgin PA-12 (white) to achieve certain colours. This is advantageous in saving on cost and materials since the virgin PA-12 is commercially available. In addition, this process allows for facile creation of brightly and/or lightly hued colours. Although we have not done this, we could create also a Black PA-12 to achieve darker colours.

Comment H: There are several locations where sentences are cutoff or missing dependent clauses, leading to difficulties in understanding the science conveyed.

The manuscript has been checked for grammatical and typographical errors.

Comment I: Figure 4B is inconsistent in formatting and display of error bars.

This has been fixed.

Comment J: The discussion of anti-biofilm assays and SEMs needs further substantiation. (j.1) The XTT assay compares reaction supernatant of each printed disk against a non-coated disk, however, there is no negative control for each printed disk not exposed to bacteria or fungi, so it is unclear how or if the XTT assay was affected by different coatings leading to indeterminate differences in assay conditions. For instance, leaching of a component (as potentially mentioned earlier in the text) may lead to absorbance or efficiency differences in the assay. Therefore, the conclusion that lack of biofilm formation is attributed solely to functionalization is unsubstantiated. (j.2) The SEMs of coated/uncoated components, specifically Figure 4D, are unclear and do not provide much, if any, additional substantiation. There are alternative methods for imaging biofilm coatings on surfaces that would provide greater evidence for lack of a visual coating or lack thereof. (j.2.1) In resolution provided, it is difficult for the reviewer to make any further observations of surface irregularities given the SEMs are unfortunately provided blurry. (j.3) While the statistical test is well described, there is no discussion of what the error bars represent within the paper.

j.1. We carried out controls against the concern raised by the reviewer. First, non-infected discs were used as a negative XTT control and these values were subtracted from the XTT absorbance obtained with the organisms. Thank you for alerting us that this wasn't mentioned in the original manuscript. We have now added this important detail in the revised Methods. Second, leaching tests performed with the functional materials showed that there was no detectable leaching into aqueous solution over a period of several weeks, more than encompassing the timescale of the assay. Second, the printed samples were cleaned and washed before experimental use including the XTT assay, to ensure the absence of contamination by any non-printed material. These controls support the interpretation that it is functionalization of the parts that affects biofilm formation according to XTT assay, and this point is now emphasised in the revised SI methods.

To the SI methods the following has been added: **First, non-infected discs were used as a negative XTT control and these values were subtracted from the XTT absorbance obtained with the organisms.**

j.2. We agree that the SEMs were not the clearest and that they did not really enhance or change any conclusions. Consequently, we have removed the SEMs in the revised manuscript and have updated Figure 4 and the related text accordingly.

j.3. The error bars represent the extremes of the varying results that were recorded for all of the samples that were tested during the XTT assays.

To the caption of figure 4 in the manuscript and S53 in the SI the following has been added: **Values shown are means from at least three biological replicates, with error bars showing standard error of the mean.**

Comment K: There is overall a lacking discussion as to the final properties of each surface coating including mentioned surface topology, roughness, hydrophobicity, and charge as well as fundamental mechanical properties expected for materials characterization. Including these findings would greatly strengthen this paper and broaden the field applicability.

The reviewer has raised a valid point that further examination of the printed parts is required, accordingly we have performed a variety of further prints and subsequent analyses.

The mechanical properties of the materials were tested and analysed as can be seen in the answer to Comment B (and comment 3 of reviewer 2).

To the SI the following was added in for the analysis of the hydrophobicity of the printed parts: Printed parts constructed from three materials and their hydrophobicity was tested via water contact angle measurements using:

- Virgin PA-12,
- PA-12 particles coated with P(IBMA-DR1MA)(Red),
- PA-12 particles coated with P(IBMA-DB3MA)(Blue).

These data revealed that there was a difference in hydrophobicity based upon the dye monomer and its structure (S64). In the sample composed of PA-12 coated with P(IBMA-DR1MA) the dye monomer is at 10 wt% (with respect to IBMA) and DR1MA is more hydrophobic, therefore the water contact angle is bigger than for the control of PA-12. Conversely the sample coated with P(IBMA-DB3MA) appears more hydrophilic, but the effect is smaller because DB3MA was used at a loading of only 2.5 wt% (with respect to IBMA) in that coating.

Sample	Water Contact Angle (°)
PA-12	85.4
Red PA-12 coated with P(IBMA-DR1MA)	107.1
Blue PA-12 coated with P(IBMA-DB3MA)	81.1

S64 – Water contact angle for the surface of printed parts.

According to the advice given by the reviewer further analysis was performed to elucidate the surface roughness and topology of the coloured printed parts and their comparison to the parts printed from the commercial PA12. The take home message is that the introduction of the coloured coating does not significantly change the surface roughness and topology after printing or after further surface processing. Note that images of the samples before and after vapour smoothing are presented in S58.

The following was added to the SI: A variety of printed samples were analysed:

- Tensile bar composed of PA-12 coated with P(IBMA-DR1MA)(Red)
- Tensile bar composed of a 50/50 mix of PA-12/PA-12 coated with P(IBMA-DR1MA)
- Tensile bar composed of an 80/20 mix of PA-12/PA-12 coated with P(IBMA-DR1MA)
- Tensile bar composed of PA-12
- Vapour Smoothed Tensile bar composed of an 80/20 mix of PA-12/PA-12 coated with P(IBMA-DR1MA)
- Vapour Smoothed Tensile bar composed of PA-12

The area measured was a single field of view (2858 mm x 2176 mm) of two different locations in every specimen, one at the tab of the sample and another at the centre (red insets in S). Alicona G5 surface texture measurements were performed (mode: 5X objective, vertical resolution of 1.50 mm, lateral resolution of 14.70 mm, z-vertical focus variation from -700 mm to +700 mm.).

The results show that the as produced “non-smoothed” printed samples had a similar surface roughness when compared to non-smoothed PA-12. Similarly, after vapour smoothing the parts made from commercial PA-12 were very similar to those printed from 80/20 mix of PA-12/PA-12 coated with P(IBMA-DR1MA) (S60-63).

Samples	Sq (μm)	Ssk	Sku	Sp (μm)	Sv (μm)	Sz (μm)	Sa (μm)
Red non-smoothed (tab)	29.75	-0.7814	4.78	96.3	193.1	289.4	22.79
Red non-smoothed (center)	36.68	-0.6145	3.882	132.9	202.5	338.4	28.95
50/50 non-smoothed (tab)	9.263	0.2374	4.35	68.7	39.81	108.5	7.137
50/50 non-smoothed (center)	10.97	0.08256	4.754	101.4	62	163.4	8.459
80/20 not-smoothed (tab)	11.69	0.5064	3.66	78.66	41.56	120.2	9.207
80/20 not-smoothed (center)	11.55	0.244	3.449	55.71	50.7	106.4	9.07
PA-12 non-smoothed (tab)	13.93	0.2435	2.879	57.66	45.4	103	11.19
PA-12 non-smoothed (center)	17.51	0.4617	3.09	70.8	45.38	116.2	13.96
80/20 smoothed (tab)	2.332	0.07847	3.643	16.12	10.6	26.72	1.825
80/20 smoothed (center)	2.206	-0.1902	4.479	13.06	13.5	26.5	1.696
PA-12 smoothed (tab)	1.815	-0.4627	4.338	10.25	12.07	22.32	1.401
PA-12 smoothed (center)	1.878	-0.6198	4.856	9.46	10.98	20.44	1.418

S60 – Results of surface roughness analysis of a variety of printed parts. Sq: root mean square height. Ssk: skewness. Sku: Kurtosis. Sp: maximum peak height. Sv: maximum pit height. Sz: Maximum height (sum of maximum peak height and maximum pit height). Sa: arithmetical mean height.

S61 – Comparison of surface roughness measurements of printed parts composed of a variety of materials.

S62 – Measurement and mapping of surface roughness of printed parts formed from. A.) Red PA-12 (PA-12 coated with P(IBMA-DR1MA)). B.) a 50/50 mix of 50% White PA-12 and 50% Red PA-12 (PA-12 coated with P(IBMA-DR1MA)). C.) an 80/20 mix of 80% White PA-12 and 20% Red PA-12 (PA-12 coated with P(IBMA-DR1MA)). D.) commercial PA-12 alone.

S63 – Measurement and mapping of surface roughness of printed parts that have been vapour smoothed A.) 80/20 mix of 80% White PA-12 and 20% Red PA-12 (PA-12 coated with P(IBMA-DR1MA)). D.) commercial PA-12.

To the manuscript the following was added: Extensive surface analysis demonstrated that parts printed from the functionalized and coloured PA-12 materials had very similar topology and surface roughness to parts printed from virgin PA-12 (S60-63).

Comments on SI:

Comment L: Please see above for some points made as to additions required to the supplemental section.

Reviewer comments have been taken into account and the same grammatical, typographical, and stylistic concerns were addressed in the SI as they were in the manuscript.

Comment M: In all $^1\text{H-NMR}$ s shown, there are several items that need to be addressed: ordering of labelled peaks, labelling of peaks on the given molecule, labelling of unintended residuals (for instance, water), presentation of integrations values and splitting, and discussion of any expected or unexpected impurities. In several of these NMRs, peaks are present that should be attributed to the molecule but are not, hydrogens that should be present are unlabeled or not visible in the spectra, etc. – this makes it difficult to substantiate that the intended molecule was synthesized and purified.

The reviewer has given us an opportunity to further clarify the analysis of our materials. To respond to this comment further peaks have been labelled for the monomers and polymers synthesized.

Further to this, the following has been added to the SI:

To the methods section, "Synthesis of Disperse Red 1 Methacrylate" the following was added: $^1\text{H NMR}$ (400 MHz, CDCl_3): δ 1.29 (3H, t, $^3\text{J} = 7.1$ Hz, CH_3), 1.98 (3H, s, $\text{C}=\text{CCH}_3$), 3.57 (2H, q, $^3\text{J} = 7.1$ Hz, NCH_2CH_3), 3.76 (2H, t, $^3\text{J} = 6.2$ Hz, $\text{NCH}_2\text{CH}_2\text{O}$), 4.40 (2H, t, $^3\text{J} = 6.2$ Hz, $\text{NCH}_2\text{CH}_2\text{O}$), 5.62 (1H, s, $\text{C}=\text{CHH}$), 6.13 (1H, s, $\text{C}=\text{CHH}$), 6.85

(2H, d, $^3J = 8.5$ Hz, 2 × ArH), 7.93 (2H, d, $^3J = 8.5$ Hz, 2 × ArH), 7.95 (2H, d, $^3J = 8.5$ Hz, 2 × ArH), 8.35 (2H, d, $^3J = 8.5$ Hz, 2 × ArH) ppm.

To the methods section, "Synthesis of Yellow 1 Methacrylate" the following was added: ^1H NMR (400 MHz, CDCl_3): δ 2.11 (3H, s, C=CCH₃), 5.82 (1H, m, C=CHH), 6.42 (1H, s, C=CHH), 7.32 (2H, m, 2 × ArH), 7.47-7.58 (3H, m, 3 × ArH), 7.94 (2H, m, 2 × ArH), 8.00 (2H, m, 2 × ArH) ppm.

To the methods section, "Synthesis of disperse blue 3 methacrylate" the following was added: ^1H NMR (400 MHz, CDCl_3): δ 1.98 (3H, s, C=CCH₃), 2.38 (3H, s, NCH₃), 3.11-3.15 (4H, m, 2 × CH₂), 5.710 (1H, s, C=CHH), 6.25 (1H, s, C=CHH), 7.28 (2H, s, 2 × ArH), 7.72 (2H, m, 2 × ArH), 8.36 (2H, m, 2 × ArH), 10.64 (2H, s, 2 × NH) ppm.

Furthermore, NMR figures S29, 32, 35, and 46-51 have been modified as recommended by the reviewer. We would like to thank the reviewer for pointing out possible improvements.

Comment O: Several axes' labels are missing. If there are multiple dependent axes in one graph, please include and label all, otherwise it is unclear what is being measured, for instance in S9.

The graphs mentioned by the reviewer are GPC chromatograms the y-axis usually carries the label: "Normalised Intensity".

To the SI "Normalised intensity" was added as the y-axis label to the following figures: S10, 15, 19, 27, 30, 33.

Comment P: Claims of DRI and UV peak overlap should be substantiated with peak max and FWHM or similar parameters to substantiate non-statistically significant differences in peak occurrence. (p.1) This same claim goes for comparing particle size. This would be much clearer if presented in a singular graph and statistical conclusions made about how much larger the particle is after each coating and statistical significance.

The dual detectors were used to show that the dye is incorporated into the polymer. The GPC peaks match and the slight delay in time between the dRI and UV polymer peaks are because of the delay in injection between the two detectors.

The particle sizing graphs obtained from LDS analysis were taken straight from the software which did not allow for graphs to be stacked, this was thought to be the most representative data, and still shows the increase in size.

No changes were made to the manuscript or SI.

Comment Q: S22 and similar graphs are hard to interpret at the current formatting. Independent and dependent axes should be blown up to better show the intended data (for instance, for S22, setting independent range to max 25 minutes and dependent range -1 to 1 (units?)).

The graphs that are mentioned are GPC chromatograms S9, 13, 17, 27, 30. The x-axis of S9, 13, and 17 has a range of 0-30 min whilst that of S17, 27, 30 has a range of 0-40 min- these have been adjusted to 0-30 min. This range was chosen as it is the full length (30 min) of the GPC experiment, hence this shows a complete representation of the samples that were analysed.

Regarding the axis labels/units, this has been addressed in Comment O of reviewer 3.

To the SI the following changes were made: Figures S27, 30, 33 were modified so that the x-axis has a range of 0-30 min.

General Changes:

To the materials and methods section of the SI the following was added as further testing was conducted on the behest of the reviewers:

Mechanical Properties Analysis:

The mechanical properties were investigated following ISO-527-2 standards (equivalent to ASTM D368) and were performed with an Instron Universal Testing system with rate of extension 1.00 mm/min, humidity 50%, and temperature 18°C.

Water Contact Angle measurements:

A Kruss DSA100 was used for water contact angle measurements of the printed part surfaces. Measurements were performed following the sessile drop method used for static contact angles.

Surface Roughness and Topology Measurements:

The area to be measured is a single field of view (2858 mm x 2176 mm) of two different locations in every specimen, one at the tab of the sample and another at the centre (red insets in Figure 1). Alicona G5 surface texture measurements were used (mode: 5X objective, vertical resolution of 1.50 mm, lateral resolution of 14.70 mm, z-vertical focus variation from -700 mm to +700 mm.). Three repeats at the same location (red rectangles, Figure 1) for every specimen under study gives a total number of 54 surface measurements under the same conditions of temperature/humidity.

Data analysis was performed on Mountains software to analyse the surface texture measurements based on ISO 25178-2 and calculate height parameters of every measurement within the scale-limited surface:

- (1) Sq: root mean square height.
- (2) Ssk: skewness.
- (3) Sku: Kurtosis.
- (4) Sp: maximum peak height.
- (5) Sv: maximum pit height.
- (6) Sz: Maximum height (sum of maximum peak height and maximum pit height).
- (7) Sa: arithmetical mean height.

Refer to ISO25178-2 Terms, definitions, and surface texture parameters for further information.

Vapour Smoothing:

Smoothing was performed via VaporFuse Surfacing

(<https://www.3dnatives.com/en/vaporfuse-surfacing-a-green-solution-for-improved-part-properties-03112020/#>)

The samples were preheated for 15 min at 145 °C. Subsequently vapour was deposited utilizing a minimum vapour pressure of 130 mbar for 11 seconds per cycle (20 cycles), after each cycle the sample was cooled to 110 °C.

REVIEWER COMMENTS

Reviewer #2 (Remarks to the Author):

The authors have made substantial improvement on the manuscript. I recommend the work to be published. However, there are still two suggestions as following.

1. Strengthened justification was provided in the response to comment one from reviewer 2. This response is recommended to be added to the main manuscript in a condensed version, since it is important for the broad audiences those are not very familiar with SLS printing.
2. Light yellow colour is not recommended to use, for neither the test of the response letter nor the figures of the main manuscript (e.g. Figure 1A and Figure 3B). It is really difficult for reader to see colours with low contrast.

Reviewer #3 (Remarks to the Author):

The authors have thoughtfully and comprehensively addressed the issues and comments raised by the reviewers. Analysis in response to comments is thorough and seeks to fully address the capabilities of the technique described in the manuscript.

Critically, we still raise the issue that the printed examples in the main text are few and lack scale bars – Figure 2 would be a great place to add more of the printed examples shown in the SI. A great way to show this would be to produce the same, complex, test shape from each of the colorants claimed in Figure 4.

It is additionally still hard to tell scale of many components as there are no relative scale bars for printed components and they are seemingly edited to a white background.

Main Text:

- Figure 1A: generic acrylate in primary mechanism is missing an oxygen.
- Figure 1A: generic polymeric coating has an extra carbon. Left bracket should be moved in for clarity. A good example is here: <https://pubs.acs.org/doi/10.1021/acsmacrolett.8b00502>

- Figure 1C, right: illustration is a surface cutaway, whereas figure such as on left, shows cutaway representative of image.
- Figure 2C,D: printed components missing scale bar.
- Figure 3C,D: printed components missing scale bar.
- Line 200 / Line 224: did the color difference have any error or variation across the surface or between samples? Replicate count / error in measurement?
- Figure 4A: All except P(IBMA) have an extra carbon in the chain.
- Line 235 – 242 makes claims regarding mechanical properties but does not state range or statistical relevance as would be of interest to AM readers.

Supplementary Information:

- Line S72 – 89: Note if commercial materials were utilized without further purification, or if any were further purified (especially so as to remove inhibitors).
- S11, S16, S21, S29, S32, S35, S46, S47, S48, S49, S50, S51: polymer contains an extra carbon.
- S45C-E, S54B, S55A-B, S56A-C, S57, S58: Scale bars missing.
- S59: would be helpful to present this data as a statistical comparison to understand which mechanical properties are statistically significantly different (especially where claimed in the main paper) as well as the error associated with each reported value.
 - o Any claim following this figure should have a statistical basis to substantiate. For example, line S970-971, is the increase statistically significant across samples? Of how much of an increase?
- S60: similar to previous note, a statistical basis to all “similar” claims should be made.

We thank the editorial team and the reviewers for their guidance and we have made a number of changes detailed below to address the specific issues and to improve the message in our manuscript and SI. As with the previous revisions changes to the manuscript are in **Red** and to the SI in **Yellow**.

REVIEWER COMMENTS

Reviewer #2 (Remarks to the Author):

The authors have made substantial improvement on the manuscript. I recommend the work to be published. However, there are still two suggestions as following.

1. Strengthened justification was provided in the response to comment one from reviewer 2. This response is recommended to be added to the main manuscript in a condensed version, since it is important for the broad audiences those are not very familiar with SLS printing.

This comment refers to "why did we use CO₂?". Looking through the abstract below you can see that we have clearly indicated why we have turned to supercritical CO₂ for this process and how we have exploited its unique properties.

"3D printing, also known as additive manufacturing, is used widely to create a vast range of 3D objects directly from a computer image. Laser Sintering (LS) or Laser Powder bed fusion (L-PBF) exploits laser processing of polymeric particles to produce 3D objects; the majority worldwide (~95%) using polyamide-12 (PA-12), the industry standard polymer, because of its ease of processing and excellent thermo-physical properties. But this necessarily constrains the functionality of the items produced; for example, they are often white or grey in appearance and must be coloured by post production processing (spray painting or vat dipping). Moreover, PA-12 printed objects show undesirable surface properties e.g. a tendency to biofoul in wet, food or medical applications. A key challenge then is to find an inexpensive route to introduce desirable functionality to PA-12, but to ensure that the materials are still printable in commercial systems. Here, we report a facile, clean and scalable approach to the modification of commercially sourced PA-12. We exploit clean supercritical carbon dioxide (scCO₂) and simple free radical polymerisations to deliver functionalised PA-12 materials that can be successfully printed using commercial apparatus. We demonstrate the potential by creating a palette of coloured PA-12 materials and we show that colour mixing could open up the opportunity to print in any colour. Additionally, we show that the same approach can be exploited to create PA-12 based objects that do not suffer biofouling in wet applications."

"Here, we report a versatile and sustainable approach to create functional powders for LS that can be printed using currently available commercial apparatus. We have demonstrated our approach by modification of commercially sourced PA-12 powder and we describe a facile process that provides each individual particle with a thin functional polymeric coating in a single step. To do this, we exploit the unique properties of supercritical CO₂ (scCO₂), which has been used previously to synthesise and process a wide range of polymers.^{19–22} The low viscosity and high diffusivity of scCO₂ allows for effective free radical polymerisation (FRP) of soluble monomers and excellent penetration of the growing functional polymer coating into the PA-12 sub-surface."

No changes made to manuscript.

2. Light yellow colour is not recommended to use, for neither the test of the response letter nor the figures of the main manuscript (e.g. Figure 1A and Figure 3B). It is really difficult for reader to see colours with low contrast.

We disagree the structure in yellow is perfectly readable and is representative of the new monomer we have created.

No changes made to manuscript.

Reviewer #3 (Remarks to the Author):

The authors have thoughtfully and comprehensively addressed the issues and comments raised by the reviewers. Analysis in response to comments is thorough and seeks to fully address the capabilities of the technique described in the manuscript.

Critically, we still raise the issue that the printed examples in the main text are few and lack scale bars – Figure 2 would be a great place to add more of the printed examples shown in the SI. A great way to show this would be to produce the same, complex, test shape from each of the colorants claimed in Figure 4.

We agree. To address this comment, we have retaken the pictures and inserted a ruler acting as a scale bar. These new pictures have been added to both the manuscript and the Supporting Information. To broaden the range of colours to respond to this comment we have introduced the "test model" built from PA-12 coated with P(IBMA-DR1MA) (Red) to the Supporting Information (S55) to complement the blue equivalent that was already present.

It is additionally still hard to tell scale of many components as there are no relative scale bars for printed components and they are seemingly edited to a white background.

See above.

Main Text:

● Figure 1A: generic acrylate in primary mechanism is missing an oxygen.

We thank the referee for spotting this. The error has now been corrected and figure 1 has been updated accordingly.

● Figure 1A: generic polymeric coating has an extra carbon. Left bracket should be moved in for clarity. A good example is here: <https://pubs.acs.org/doi/10.1021/acsmacrolett.8b00502>

This error has been fixed and figure 1 has been updated accordingly, seen in Comment 1 of reviewer 3.

- **Figure 1C, right: illustration is a surface cutaway, whereas figure such as on left, shows cutaway representative of image.**

The two cutaway images are now identical and are representative of the SEM image.

- **Figure 2C,D: printed components missing scale bar.**

We thank the reviewer for this comment as it gives us the chance to better illustrate the potential of our materials. Pictures have been retaken with a ruler as a scale bar and added to the figures accordingly.

- **Figure 3C,D: printed components missing scale bar.**

The diagram is already complex. But to be clear we have now added extra pictures with a scale bar ruler to clearly demonstrate to the reader the size of the object. See Supporting Information S58.

S58 – Images of non-smoothed and post vapour smoothed printed parts composed of an 80/20 mix of virgin PA-12 and PA-12 coated with P(IBMA-DB3MA) showing an improvement in surface finish. A.) Top view of non-smoothed part. B.) Bottom view of non-smoothed part. C.) Top view of smoothed part. D.) Bottom view of smoothed part.

● **Line 200 / Line 224: did the color difference have any error or variation across the surface or between samples? Replicate count / error in measurement?**

The reviewer makes a good point and we have now demonstrated that the colour differences would be imperceptible to the naked eye. In detail, the colour of the printed parts was measured in triplicate for the base colours in different sections of the built part using a NIX colour sensor. This device takes an image of a section of the part (~1 cm). For example in the square shaped parts seen in Figure two of the manuscript the colour was measured on the back and front surfaces of the part. This device takes an image of a section of the part (~1 cm). An example of this can be seen below (Figures 1-3):

Figure 1 – NIX color analysis results for yellow printed part 1

Figure 2 – NIX color analysis results for yellow printed part 2

Figure 3 – NIX color analysis results for yellow printed part 3

Utilizing the color difference equation for ΔE (Colour Difference Method – Delta E CMC SI 1) it can be seen that the difference between the colors is less than 1, therefore the difference in color is not perceptible to the human eye.

- **Figure 4A: All except P(IBMA) have an extra carbon in the chain.**

This error has been fixed and figure 4 has been updated accordingly.

- **Line 235 – 242 makes claims regarding mechanical properties but does not state range or statistical relevance as would be of interest to AM readers.**

We can see that use of the word 'similar' to describe the mechanical properties of the printed parts is not ideal. To overcome this, further analysis has been added to the manuscript and supporting information to enhance the comparison between the control (PA-12) and novel coated materials.

To the manuscript the following statement has been removed: Coloured and functional PA-12 based materials can be processed easily by SLS to produce 3-D objects with very good mechanical properties. In fact, the 80/20 % mix of commercial PA-12 and PA-12 coated with P(IBMA-DR1MA) showed an increased elastic region and higher yield stress compared to the materials printed under identical conditions using the commercial PA-12 materials (S59).

And replaced with: The mechanical properties of the control (virgin PA-12) and the coloured and functional PA-12 based materials were analysed utilizing ISO-527-2 standards in triplicate (S59-60). Compared to the control parts composed of virgin PA-12, a mix of commercial PA-12 and PA-12 coated with P(IBMA-DR1MA)(80/20wt%) had a mean tensile stress at maximum load of 39.61 MPa, over 97% of that measured for the control PA-12 (S59-60). So, we can be confident that our introduction of colour does not have a detrimental effect upon the 3D printed parts.

The range in the measured tensile stress at maximum load between the printed parts for the 80/20 mixed samples was ~2% and that of virgin PA-12 was ~1% (S59-60) this suggests that there is further optimisation of print parameters required, and that the P(IBMA-DR1MA) is having some impact on the energy absorption characteristics of prepared material that requires further investigation.

From the mechanical property data, the coloured and functional parts have been printed successfully via laser sintering processes with minimal changes to manufacturing process parameters.

To the Supporting information the following has been added:

	PA-12 1	PA-12 2	PA-12 3	80/20 mix 1	80/20 mix 2	80/20 mix 3
Tensile stress at maximum load (MPa)	40.76	40.6	40.96	39.77	39.11	39.95
Tensile strain at break (%)	2.29	1.72	2.22	1.67	1.35	1.61

S60 – Summary of individual mechanical testing results (in triplicate) for parts printed with commercial PA-12 and parts printed with an 80/20 mix of commercial PA-12 and PA-12 coated with P(IBMA-DR1MA). Parts tested with ISO-527-2 protocols.

Supplementary Information:

- **Line S72 – 89: Note if commercial materials were utilized without further purification, or if any were further purified (especially so as to remove inhibitors).**

We would like to thank the reviewer for this comment. We used commercial materials without any further purification.

No Changes to the manuscript or supporting information

- **S11, S16, S21, S29, S32, S35, S46, S47, S48, S49, S50, S51: polymer contains an extra carbon.**

This error has been fixed and figure 1 has been updated accordingly.

- **S45C-E, S54B, S55A-B, S56A-C, S57, S58: Scale bars missing.**

We thank the reviewer for this comment as it gives us the chance to better illustrate the potential of our materials. Pictures have been retaken with a ruler as a scale bar and added or replaced in the figures (S54, S55, S56, S57, S58) accordingly. An example can be seen below.

S57 – Images of lattice type structure built from PA-12 coated with P(IBMA-DR1MA). A.) Front end view. B.) Top view C.) Side end view D.) Diagonal view.

- **S59:** would be helpful to present this data as a statistical comparison to understand which mechanical properties are statistically significantly different (especially where claimed in the main paper) as well as the error associated with each reported value.
 - o Any claim following this figure should have a statistical basis to substantiate. For example, line S970-971, is the increase statistically significant across samples? Of how much of an increase?

We thank the reviewer. We would like to clarify that for this study we are not claiming that our materials have superior mechanical properties when compared to commercial virgin PA-12, but instead that the coating process does not significantly positively or negatively impact the mechanical properties of the SLS printed parts.

This has been clarified and covered above.

- **S60:** similar to previous note, a statistical basis to all “similar” claims should be made.

We would like to thank the reviewer for the opportunity to add further detail to our study. To address this point further clarification of the statistical analysis was performed to compare the topology of the 80/20 mix and commercial virgin PA-12.

*We have clarified this point with the following text: **In addition, extensive surface analysis revealed that the coloured PA-12 materials and the control are in the same range of values in terms of surface roughness (S61-68).***

*And we have removed from the manuscript this original text: **In addition, extensive surface analysis demonstrated that parts printed from the functionalized and coloured PA-12 materials had very similar topology and surface roughness to parts printed from virgin PA-12 (S60-63).***

*To reinforce these same points we have added to the Supporting Information the following text and tables: **The results show that the 'non-smoothed' 80/20 mix printed parts without post processing yielded components that were smoother than their control counterparts from virgin PA-12 (S62-63), as the Sq and Sa values for the 'non-smoothed' 80/20 mix printed parts are lower than the control and the Sp and Sv are in the same range. Again, this suggests that the 80/20 mix material is highly suited to laser sintering processes.***

	Mean of 80/20 mix 'non-smoothed' tab	Mean of 80/20 mix 'non-smoothed' centre	Mean of total 80/20 mix 'non-smoothed'	Range
Sq (µm)	11.68	11.54	11.61	11.54-11.69
Sp (µm)	78.68	56.21	67.45	55.71-78.73
Sv (µm)	41.4	50.73	46.07	41.28-50.99
Sa (µm)	9.19	9.062	9.13	9.06-9.21

S62 – Topology and surface roughness of “non-smoothed” printed samples composed of an 80/20 mix of commercial virgin PA-12 and PA-12 coated with P(IBMA-DR1MA).

	Mean of PA-12 'non-smoothed' tab	Mean of PA-12 'non-smoothed' centre	Mean of total PA-12 'non-smoothed'	Range
Sq (µm)	13.93	17.5	15.72	13.91-17.51

Sp (µm)	58.49	70.61	64.55	57.60-70.80
Sv (µm)	45.15	45.51	45.33	44.96-45.85
Sa (µm)	11.19	13.95	12.57	11.17-13.96

S63 – Topology and surface roughness of "non-smoothed" printed samples composed of control PA-12.

Carrying out the same analysis for the 'smoothed' 80/20 mix printed samples. These are slightly less smooth compared to the control PA-12 samples as Sq, Sp, Sv, Sa values are higher than those of the control (S64-65). The increase in surface roughness between the 80/20 mix and the commercial virgin PA-12 after smoothing could be because the same "smoothing" procedure was utilized for both materials. This 'smoothing' procedure has been industrially optimized for commercial PA-12; therefore, it is probable that with "smoothing" procedure optimization for the 80/20 mix the differences would become smaller.

	Mean of 80/20 mix 'smoothed' tab	Mean of 80/20 mix 'smoothed' centre	Mean of total 80/20 mix 'smoothed' '	Range
Sq (µm)	2.33	2.2	2.27	2.20-2.33
Sp (µm)	16.23	13.32	14.78	13.00-16.38
Sv (µm)	10.67	13.45	12.06	10.60-13.50
Sa (µm)	1.83	1.7	1.77	1.67-1.83

S64 – Topology and surface roughness of "smoothed" printed samples composed of an 80/20 mix of commercial virgin PA-12 and PA-12 coated with P(IBMA-DR1MA).

	Mean of PA-12 'smoothed' tab	Mean of PA-12 'smoothed' centre	Mean of total PA-12 'smoothed' '	Range
Sq (µm)	1.82	1.88	1.85	1.81-1.88
Sp (µm)	9.93	9.48	9.71	9.46-10.25
Sv (µm)	12.26	11.04	11.65	10.98-12.41
Sa (µm)	1.4	1.42	1.41	1.40-1.42

S65 – Topology and surface roughness of "smoothed" printed samples composed of a control PA-12.

Having made the above changes in the Supporting Information the following text was removed: The results show that the as produced "non-smoothed" printed samples had a similar surface roughness when compared to non-smoothed PA-12. Similarly, after vapour smoothing the parts made from commercial PA-12 were very similar to those printed from 80/20 mix of PA-12/PA-12 coated with P(IBMA-DR1MA) (S60-63).